# Broad modulus range nanomechanical mapping by magnetic-drive soft probes

Xianghe Meng[1], Hao Zhang[1], Jianmin Song[1], Xinjian Fan[1], Lining Sun[1] & Hui Xie[1]

Stiffness matching between the probe and deformed portion of the sample in piezo-drive peak force modulation atomic force microscopy (AFM) limits the modulus measurement range of single probes. Here we develop a magnetic drive peak force modulation AFM to broaden the dynamic range of the probe with direct cantilever excitation. This approach not only successfully drives the softest commercial probe (6 pN nm$^{-1}$) for mapping extremely soft samples in liquid but also provides an indentation force of hundreds of nanonewtons for stiff samples with a soft probe. Features of direct measurements of the indentation force and depth can unify the elastic modulus range up to four orders of magnitude, from 1 kPa to 10 MPa (in liquid) and 1 MPa to 20 GPa (in air or liquid) using a single probe. This approach can be particularly useful for analysing heterogeneous samples with large elastic modulus variations in multi-environments.

[1] State Key Laboratory of Robotics and Systems, Harbin Institute of Technology, 2 Yikuang, Harbin 150080, China. Correspondence and requests for materials should be addressed to H.X. (email: xiehui@hit.edu.cn)

Atomic force microscopy (AFM)-based force spectroscopy quantitatively measures the local properties of materials and intermaterial interactions with nanoscale spatial and picoNewton force resolution. The quasistatic loading technique was developed as an early effort to study the distance dependence between the force and the probe-sample separation[1], and it has expanded our understanding of the nanomechanics of a large variety of materials, such as semiconductors[2], metals[3], polymers[4], biomaterials[5], and emerging nanomaterials[6]. This technique has inspired two representative force spectroscopies: single-molecule force spectroscopy[7,8] and single-cell force spectroscopy[9]. The former is a powerful tool for characterizing intermolecular and intramolecular forces[10], and it has been widely used to study molecular mechanisms[11], structures[12,13], and dynamics[14]. The latter is a specialized method for quantifying intercellular adhesion, cell to extracellular matrix adhesion[15,16], and single adhesion–receptor–ligand interactions[17].

The capability of multiparametric imaging with nanoscale spatial resolution has brought force spectroscopy to a new exciting stage[18]. In contrast to the conventional way of measuring force-distance (FD) curves, in which only selected points are measured, two main imaging modes have been developed for quantitatively mapping the local properties of a sample: force-volume imaging[19] and jumping[20] or peak force tapping[21]. The peak force tapping and its derivates enable faster mapping of the nanomechanical properties at the rate of conventional topographic AFM imaging[22,23] and it has been extensively applied for the nanomechanical quantification of complex cellular and biomolecular systems[24–26] and soft[27] and rigid[28] materials under liquid or ambient environments.

In the abovementioned force spectroscopy methods, the indentation depth is measured indirectly and generally converted from the deflection of the probe and displacement of the piezoelectric translator[29]. Uncertainties from the AFM system will degrade the measurement accuracy, especially when quantifying the forces. For instance, for a very stiff cantilever, the loading force can result in excessive indentation making it uninterpretable by mechanics modeling[30–32]. In contrast, indentation of hard materials with a soft probe can exceed the linear range of the force measurement as the cantilever bends too much trying to achieve sufficient indentation. The hysteresis of the piezoelectric translator and vibration coupling in the peak force tapping mode are other constraints. The recommended solution is to choose a probe with a spring constant close to that of the sample[18,22,23]. Although an appropriate probe can be found after several attempts of using a sample with unknown elastic properties, it remains a challenge to use the same probe for a heterogeneous surface with large variations in its elastic modulus of one or two orders of magnitude[22,23].

In this work, a magnetic drive peak force modulation AFM is developed to break the limit of the nanomechanical measurement range of probes. In this approach, instead of vibrating the probe holder, only the cantilever beam is sinusoidally oscillated by the magnetic torque at selected off-resonance frequencies. The vibration coupling noise is minimized due to the negligible total mass of the cantilever beam and the attached magnetic bead (Ø3–15 µm). With this method, we can directly monitor the position of the tip in real time using the cantilever response during the entire measurement process. As a result, both the force and the indentation depth can be directly measured. The optimized magnetic drive system provides equivalent drive forces up to several hundreds of nanoNewtons, and the force measurement range of AFM is maximized using a nonlinear force calibration method[33]. Therefore, it is possible to use the same probe to measure the elastic modulus over four orders of magnitude, which is even wider than that of bimodal[34] and torsional harmonic tapping AFMs[35]. Our experiments demonstrate that the proposed method covers a wide range of elastic moduli from 1 kPa to 20 GPa. This is achieved by using only two soft probes with nominal spring constants of 0.006 N m$^{-1}$ (in liquid) and 2 N m$^{-1}$ (in air or liquid) to unify multiple discrete modulus ranges from 1 kPa to 10 MPa and 1 MPa to 20 GPa, respectively. Thus, it is referred as Broad Modulus Range Nanomechanical Mapping (BMR NM).

## Results

**System design.** The experimental setup for BMR NM consists of the most advantageous parts of its predecessors (see Supplementary Note 1), combined with our approaches. Two main components, similar to prior magnetic-actuation setups[36–38] can be seen in Fig. 1a–i: first, a compact solenoid mounted underneath or on the sample plate is used to produce an AC magnetic field on the z–axis with an enhanced magnetic drive strength ($\mathbf{B_z}$) up to one hundred mT and a frequency of >10 kHz. Second, a ferromagnetic bead (Ø3–15 µm) is attached on the backside of the cantilever (exactly over the probe tip), and the bead is magnetized with a pulse magnetic field of ~5 T along the longitudinal axis of the cantilever (see Supplementary Note 7). For compatibility with the custom AFM system, the compact solenoid is embedded in a metal case with an optimized size of an Ø8 mm diameter and 2 mm height. The attached magnetic bead is subjected to a torque $\boldsymbol{\tau} = \mathbf{m} \times \mathbf{B_z}$ ($\mathbf{m}$ is the magnetic moment), bending the cantilever. By varying the magnitude of frequency and the drive current through the solenoid, the off-resonance oscillation frequency and amplitude of the cantilever are precisely controlled. The Fig. 1a–ii shows a schematic of the free oscillation probe with an amplitude $A_v$ that corresponds to the voltage output of the position-sensitive detector (PSD). With this drive method, intermittent indentation occurs when the sample enters the motion scope of the probe (Fig. 1a–iii). In this case, the probe bending during contact (region II) is modulated due to the tip-sample interaction forces, and a transient vibration signal caused by the adhesion force is superimposed during out-of-contact (region I) phase when the tip pulls off from the sample surface.

Figure 1b shows scanning electron microscope (SEM) images of the two probes used in our experimental study. The Fig. 1b–i (Probe I) is a commonly used (B-lever of HQ:NSC36/No Al, nominal spring constant of 2 N m$^{-1}$) for NM in air and liquid, while the Fig. 1b–ii (Probe II) is the softest commercial cantilever available (B-lever of Olympus Biolever BLRC150 VB, nominal spring constant of 0.006 N m$^{-1}$) for NM in liquid. Microassembly method was used to glue (DP760 epoxy adhesive, see Supplementary Note 7) magnetic beads (MQP-S-11-9, Magnequench, Singapore) with diameters of Ø11.4 µm and Ø3.8 µm precisely over the tips of Probe I and II, respectively. The size of the magnetic beads was carefully selected by considering the strength of the magnetic drive force and added-mass influence on the probe dynamics.

Figure 1c shows the block diagram of the data processing and peak force control for NM. By comparing previously recorded free oscillation signals ($U_M(t)$) (dark blue curve) with the probe response caused by the intermittent indentation ($U(t)$) (pink curve), the contact force signal ($F_{ts}(t)$) (dark orange curve) can be directly obtained as the difference between these two signals, considering the probe dynamics and provided that the laser spot was located at two-thirds of the cantilever length (Fig. 1c inset). By multiplying the coefficient $\gamma$ with $U(t)$, the tip position $z(x_{tip},t)$ was obtained (See the Methods section). More importantly, as shown in partially enlarged detail (Fig. 1c inset), the instantaneous indentation depth ($\delta$) can be measured directly from the probe response at region II. A force-distance curve is

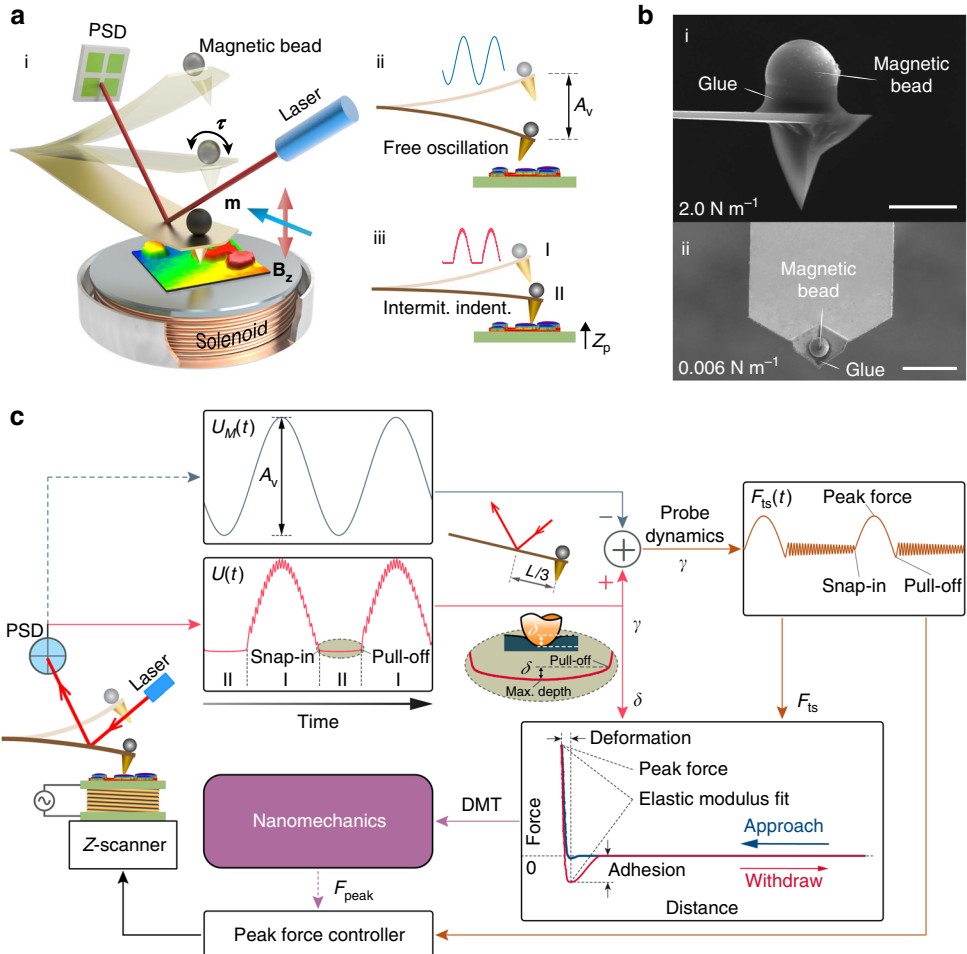

**Fig. 1** Scheme of magnetic drive peak force modulation AFM. **a** (i) Design of the experimental setup in which a magnetic bead probe is oscillated at an off-resonance frequency by a magnetic torque $\tau$ in an AC magnetic field ($B_z$) generated by the solenoid underneath. (ii) The probe oscillates near the sample surface with a free amplitude of $A_v$ and (iii) intermittently indents the sample through the translation of the $Z$ piezo-scanner. **b** SEM images of the probes with spring constants of $2\,\text{N}\,\text{m}^{-1}$ (i) and $0.006\,\text{N}\,\text{m}^{-1}$ (ii) attached with $\varnothing11.4\,\mu\text{m}$ and $\varnothing3.8\,\mu\text{m}$ magnetic beads, respectively. Scale bar, 10 μm. **c** Block diagram depicting the data processing flow and feedback control of the peak force. The tip-sample interactive response ($F_{ts}$) (dark orange) and the indentation depth ($\delta$) can be directly obtained from probe responses $U(t)$ with (pink) and $U_M(t)$ without (dark blue) contact on the sample when the laser spot is located at two-thirds of the length ($L$) of the probe. The nanomechanical properties can then be quantified using the acquired force-distance data

subsequently obtained for quantitative measurement of the nanomechanical properties. The maximum $F_{ts}$, namely, the peak force ($F_{peak}$), is modulated by the peak force controller. While mapping heterogeneous surfaces with large elastic modulus variations, $\delta$ is required to be well confined within reasonable bounds by regulating the peak force setpoint according to the immediate nanomechanical measurement. These features enabled us to use a single probe to map surfaces with an elastic modulus difference over several orders of magnitude.

**Magnetic drive strength**. Figure 2a shows the magnetic drive strength of the developed microscope that is measured with the silicon probe with an attached $\varnothing11.4\,\mu\text{m}$ ferromagnetic bead (Fig. 1b–i). The plots of the experimental data were recorded while scanning the probe over the magnetic coil with a distance of 1 mm. A voltage amplifier was used to energize the coil to adopt common peak force tapping frequencies of 1 and 2 kHz using different drive currents ($A_c$) ranging from 3.4–26.7 mA and 2.4–29.5 mA, respectively. The equivalent force amplitude ($A_f$) produced by the magnetic torque ($\tau$) is determined by $A_f = k_d\gamma A_v$ (where $k_d$ is the probe dynamic spring constant and $\gamma$ is the optical lever sensitivity). The magnitude of $A_f$ (nearly linear with

that of $A_c$) can be regulated freely up to 425 nN or greater (not shown here). The magnetic drive strength can be used to measure the elastic moduli up to 20 GPa or higher, using Derjaguin-Muller-Toporov (DMT) theory (3 nm indentation depth and 10 nm tip radius). For example, at the 2 kHz drive frequency, $A_f = 50$ nN when $A_c = 2.4$ mA for an elastic modulus less than 2 GPa, $A_f = 230$ nN when $A_c = 16.5$ mA for an elastic modulus up to 10 GPa, and $A_f = 420$ nN ($A_c = 28.6$ mA) can be used to quantify the elastic modulus up to 20 GPa. In addition, the heating effect of the coil should be well controlled to accurately measure the nanomechanical properties of temperature sensitive materials, e.g., polymers. The variations in the coil temperature are very small in an open environment because the thermal loss ($P_{loss}$) is limited to 0.03 W (a coil resistance 330 Ω) when measuring elastic moduli up to 5 GPa, which covers the testing requirements of most polymers. For the NM of soft samples in liquid, a magnetic bead with a size of $\varnothing3.8\,\mu\text{m}$ (Fig. 1b–ii) is used to drive the 0.006 $\text{N}\,\text{m}^{-1}$ probe. Considering the liquid damping and the inertial effects, the magnetic drive force at 250 Hz is calibrated (see Supplementary Note 3). The drive current $A_c = 6.9$ mA is applied to produce a drive force of 1.54 nN that is sufficient for the mapping of elastic moduli ranging from 1 kPa to 10 MPa in liquid with small adhesion.

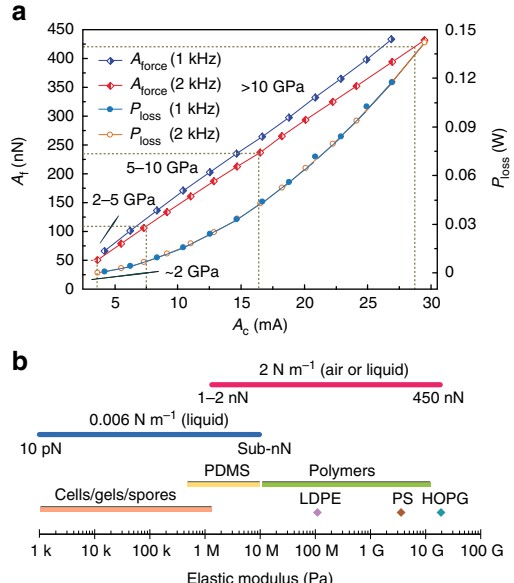

**Fig. 2** Magnetic drive strength and modulus range of the NM technique. **a** Calibrated magnetic drive force ($A_f$) and $P_{loss}$ when the probe was oscillated at 1 and 2 kHz with different drive currents ($A_c$). **b** Elastic moduli of various common matter ranges from 1 kPa to 20 GPa are unified into two modulus spectra using only two probes with spring constants of 0.006 N m$^{-1}$ and 2 N m$^{-1}$

Benefiting from the direct measurements of the indentation depth and enhanced magnetic drive strength, a broad range of elastic moduli can be mapped using the same probe, as described in Fig. 2b. The elastic modulus range was determined by the controllable peak force. For instance, Probe I provided controllable peak forces ranging from 0.1 nN to more than 500 nN for quantifying the elastic moduli from 1 MPa to 20 GPa (close to the measurement limit of the silicon probe) with a reasonable indentation depth under a controlled environment. This range covers from frequently used polymers, highly oriented pyrolytic graphite (HOPG), connective tissue, woods, bone and biological fibrils/fibers/proteins. Furthermore, the magnetic drive successfully resists strong liquid damping and perfectly drives Probe II while nanomechanically mapping elastic moduli from 1 kPa to 10 MPa in liquid; this range covers almost all soft and biological polymers, cells (eukaryotes and prokaryotes), gels and spores etc. Compared to some of the most commonly used probes' discrete working range[22,23], the probes in this work can cover the whole spectrum with two unified spectra.

To demonstrate the capability, robustness and adaptability of the presented BMR NM technique, different samples with elastic moduli ranging over four orders of magnitude were tested using the same probe both in liquid and in air.

**BMR NM in air.** The nanomechanical properties of two samples of polydimethylsiloxane (PDMS) (elastic modulus of 2.5 MPa, PDMS-SOFT-1-12 M, Bruker Nano Inc.) and HOPG (nominal elastic modulus of 18 GPa, HOPG-12 M, Bruker Nano Inc.) were accurately mapped using Probe I in air (25 °C). To minimize the capillary effect, the AFM system was placed inside a sealed chamber with relative humility below 5%. The magnetic drive frequency was set as 2 kHz. The drive amplitude of the probe was set as 100 nm while scanning the soft PDMS sample. It was set as 600 nm while scanning the stiffer HOPG sample to provide a sufficient indentation force of over several hundreds of nano-Newtons. Single force–distance curve tests were performed prior

to scanning to determine the peak force setpoint, which was 0.5 nN for the PDMS and 400 nN for the HOPG, to produce an effective indentation depth of approximately three nanometers. DMT contact mechanics theory was used to estimate the elastic modulus. The mechanics of the probe material were considered in the elastic modulus calculation of the stiff HOPG using the following equation:

$$E_{HOPG} = \frac{(1 - \nu_{HOPG}^2)E_r E_{si}}{E_{si} - E_r(1 - \nu_{si}^2)} \qquad (1)$$

where $E_{si} = 160$ GPa and $\nu_{si} = 0.278$ indicate the elastic modulus and Poisson's ratio of the silicon, respectively, and $E_r$ and $\nu_{HOPG} = 0.24$ are the measured reduced elastic modulus and Poisson's ratio of the HOPG, respectively.

The AFM topography of the PDMS sample shows a granular form as can be seen in Fig. 3a with a grain diameter of ~20 nm (Fig. 3f). Maps of the adhesion force, indentation depth, contact stiffness, and reduced elastic modulus were simultaneously determined together with the topography, and they indicated the consistency in the granularity of the surface (Fig. 3b–e). The adhesion force was measured as 1.73 ± 0.22 nN (Fig. 3g), and the indentation depth was controlled at 29.9 ± 1.9 nm (Fig. 3h). The histogram in Fig. 3i shows the value of the contact stiffness centered at 0.118 N m$^{-1}$, which is comparable to the probe stiffness. The estimated reduced elastic modulus was measured to be 3.15 ± 0.36 MPa, as shown in the histogram of Fig. 3j, which resulted in an elastic modulus of 2.42 ± 0.27 MPa with a Poisson's ratio of 0.48. This value is very close to the nominal value of the tested PDMS.

The AFM topography in Fig. 3k shows the nanomechanical maps of HOPG, scanned over a multi-layered area (Fig. 3p). The adhesion force, indentation depth, contact stiffness and elastic modulus were simultaneously mapped with the typography, which exhibited uniform contrast, except for some features at the layer boundaries (Fig. 3l–o). The peak force was set as 400 nN, together with the adhesion force of 16.95 ± 1.05 nN (Fig. 3q), to obtain a sufficient indentation depth of 3.16 ± 0.36 nm (Fig. 3r) for the subsequent nanomechanics calculations. The histogram in Fig. 3s shows the contact stiffness of 162.32 ± 10.11 N m$^{-1}$, which was close to two orders of the magnitude of the probe stiffness, and was well mapped with satisfactory accuracy and image contrast. Similarly, the reduced elastic modulus of the HOPG surface was measured as 17.53 ± 1.14 GPa (Fig. 3t), which resulted in an elastic modulus of 18.38 ± 1.33 GPa (Poisson's ratio 0.24) that is consistent with the nominal values (18 GPa) of the HOPG.

In addition, the capability of liquid BMR NM with Probe I was demonstrated by scanning the Finegoldia Magna bacteria in deionized water (see Supplementary Note 4).

**BMR NM in liquid.** The capability of liquid NM operation was demonstrated by scanning two samples of polyacrylamide gel (PA Gel) and polydimethylsiloxane (PDMS) (elastic modulus 3.5 MPa, PDMS-SOFT-2-12 M, Bruker Nano Inc.) in deionized water (25 °C) using Probe II. The magnetic drive frequency is set as 250 Hz, and the probe's dynamics was fully considered because of its largely reduced first resonant frequency (1.72 kHz) in water. The drive amplitude was set as 200 nm for the soft PA Gel sample and 250 nm for the PDMS sample. Prior to scanning, a single force–distance curve test was performed to determine the peak force setpoint, which was optimized at 35 pN for the PA Gel sample and 0.8 nN for the PDMS sample, to obtain the appropriate indentation depth while achieving stable control. Particularly, the elastic modulus was estimated using Hertz model for the PA Gel and PDMS samples because the adhesion was not

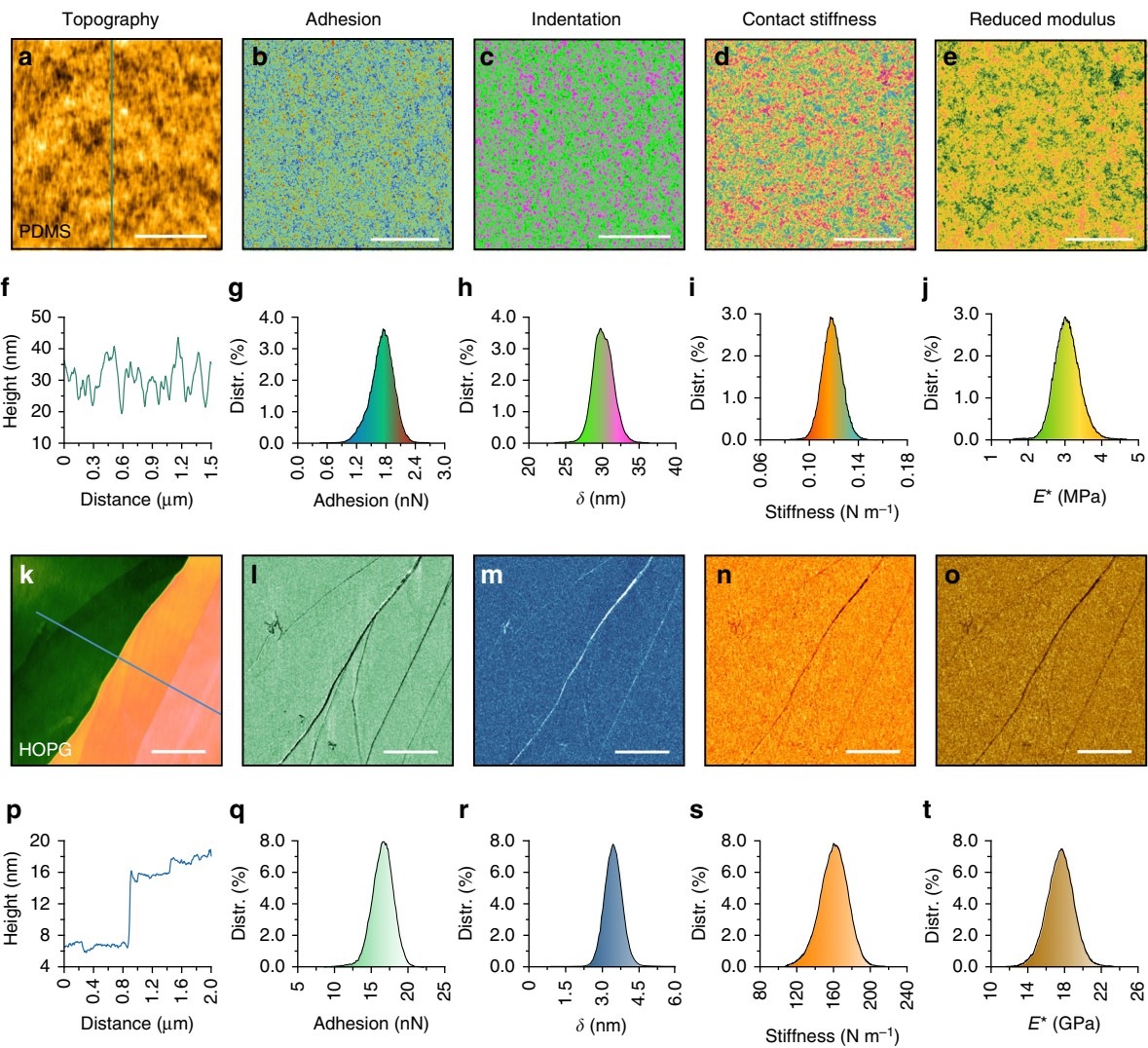

**Fig. 3** Nanomechanical maps of the PDMS and HOPG surfaces using Probe I in air. **a–e** PDMS topography image and the maps of adhesion, indentation, contact stiffness and reduced elastic modulus. **f** Cross-section along the line shown in **a**. **g–j** Corresponding histograms obtained from **b** to **e**. **k–o** HOPG topography image and maps of adhesion, indentation, contact stiffness and elastic modulus. **p** Cross-section along the line shown in **k**. **q–t** Corresponding histograms obtained from **l** to **o**. Scale bar, 500 nm

observed. The magnetic drive force was recorded as the probe tip became very close to the sample surface to remove the hysteresis effect caused by hydrodynamic damping in the liquid media.

The AFM topography of the PA Gel is shown in Fig. 4a, and the corresponding cross section described in Fig. 4e indicates that the PA Gel had an irregular self-affine structure with fluctuations of ~30 nm. These features were correlated in nanomechanical maps as shown in Fig. 4b–d. Figure 4f–h shows the corresponding histograms. The indentation depth ($\delta$) was controlled at $64.2 \pm 9.5$ nm (Fig. 4f). The averaged contact stiffness at the full indentation depth was statistically calculated as $0.00186 \pm 0.00014$ N m$^{-1}$ (Fig. 4g). Although the indentation depth exceeded the radius of the probe, Hertzian contact theory was used to estimate the elastic modulus because the special half-pyramid tip shape of the Probe II prevented the application of Sneddon contact theory. A reasonable reduced elastic modulus was calculated as $8.5 \pm 0.61$ kPa (Fig. 4h), which resulted in an elastic modulus of $6.37 \pm 0.46$ kPa with a Poisson's ratio of 0.5. This value was supported by the measurements obtained while changing the relative concentration of acrylamide to bisacrylamide[39].

Figure 4i shows the AFM topography of the PDMS sample. The cross section depicted in Fig. 4m shows that the PDMS

surface featured grains with a diameter of approximately 15 nm. The contrast observed in the nanomechanical maps (Fig. 4j–l) indicated consistencies in the granularity of the surface observed in the topography image. As described in the histograms of Fig. 4n–o, the indentation depth was controlled at $7.93 \pm 0.61$ nm, and the mean value of the contact stiffness was measured as $0.153 \pm 0.007$ N m$^{-1}$. The value of the reduced elastic modulus was centered at 4.41 MPa with a standard deviation of 0.25 MPa (Fig. 4p), which resulted in an elastic modulus of $3.39 \pm 0.19$ MPa with a Poisson's ratio of 0.48. This elastic modulus value is very close to the nominal value (3.5 MPa) of the tested PDMS.

**BMR NM on a heterogeneous surface**. To demonstrate the capability of the proposed method for measuring heterogeneous surfaces with large variations in the elastic modulus, a blend composed of polystyrene (PS) and a polyolefin elastomer (LDPE) (PS-LDPE-12 M, Bruker Nano Inc.) were measured using Probe I. The nominal elastic moduli of the PS and LDPE regions are 2 and 0.1 GPa, respectively. The measurements were performed using Probe I under the same experimental conditions in air. In Fig. 5a, the AFM topography shows that the blend was composed of

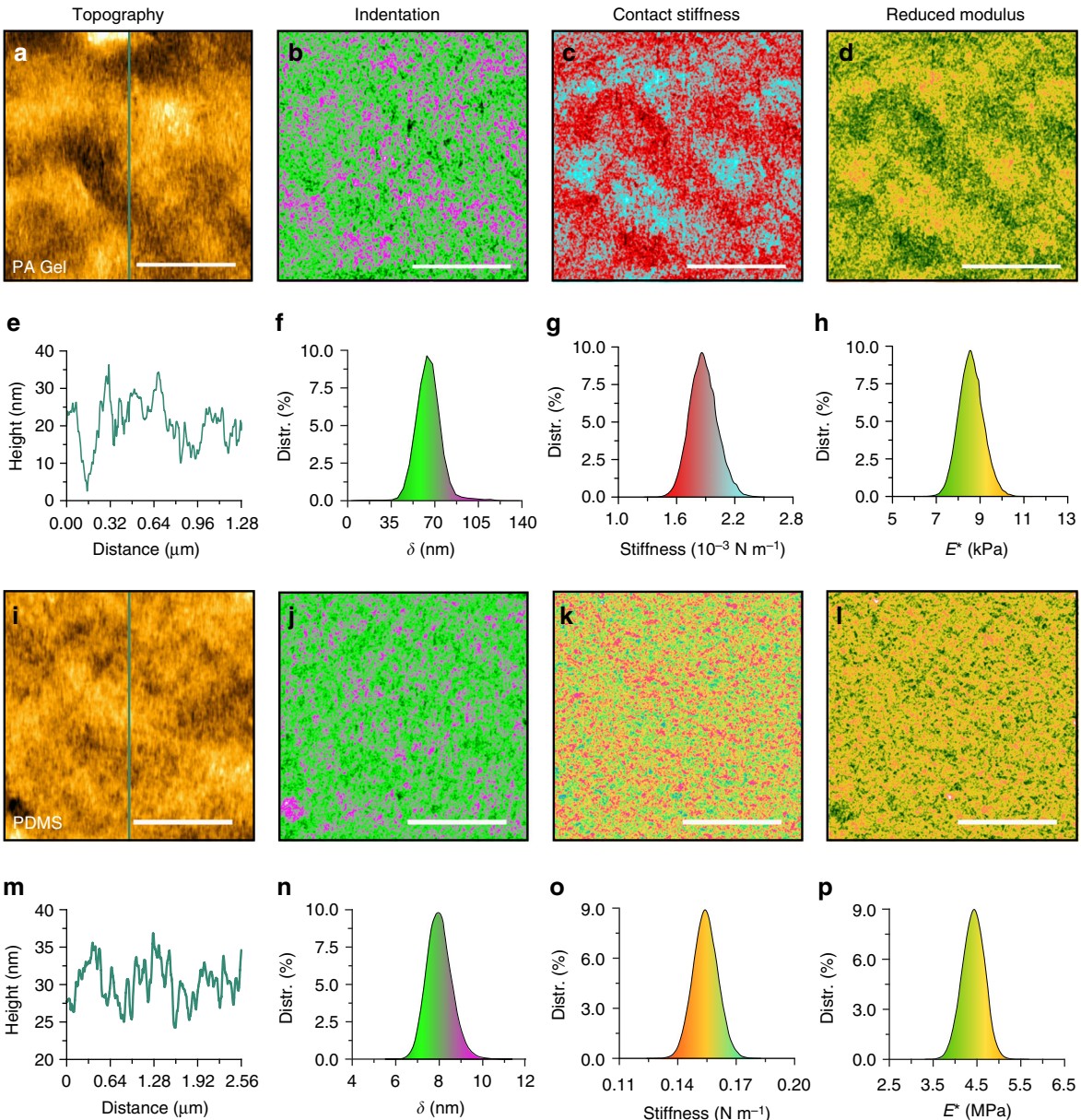

**Fig. 4** Nanomechanical maps of the hydrogel and PDMS using Probe II in liquid. **a–d** PA Gel topography image, and the maps of indentation, contact stiffness and reduced elastic modulus. **e** Cross-section along the line shown in **a**. **f–h** Corresponding histograms obtained from **b** to **d**. **i–l** PDMS topography image, and the maps of indentation, contact stiffness and reduced elastic modulus. **m** Cross-section along the line shown in **i**. **n–p** Corresponding histograms obtained from **j–l**. The scale bars are 500 nm for the PA Gel and 1 μm for the PDMS sample

circular LDPE islands embedded in a PS matrix covered with tiny grains that were several nanometers high. The diameters of the LDPE islands were 0.5–1.0 μm and ~45–50 nm high (cross section in Fig. 5d). The blend characteristics clearly appeared in the adhesion map (Fig. 5b) with different mean values of 9.4 nN (LDPE) and 11.9 nN (PS), as shown in the histogram in Fig. 5e.

From the regulation law described in eq. 11 (see the Methods section), the maximum indention force ($F_{peak}$) was confined to approximately 45.3 nN ($N = 4$ and $\delta_t = 3.0$ nm) on the PS region, while it was ~2 nN ($N = 4$ and $\delta_t = 6.5$ nm) on the LDPE region, as shown in Fig. 5f (map of the indentation force) and Fig. 5j (indentation force profile). According to the DMT theory, the total indentation forces ($F_{DMT}$) ($F = F_{peak} + F_{adh}$) applied on the LDPE and PS regions were ~11.4 nN and 57.2 nN, respectively. Consequently, the regulated indentation forces yielded appropriate indentation depths of 7.75 nm and 3.45 nm on the LDPE and PS regions (Fig. 5g, k), respectively.

The maps of the mechanical properties clearly show a pattern that perfectly agreed the morphology. The stiffness map (Fig. 5h) shows two mean values in the histograms of Fig. 5l, one centered at 1.12 N m$^{-1}$ (LDPE) and the other at 17.85 N m$^{-1}$ (PS), which were calculated by fitting 90% of the retracted path of the FD curve using the least squares method. The map of the reduced elastic modulus in Fig. 5i shows that the circular regions of LDPE had an elastic modulus of 101.2 ± 2.0 MPa (Poisson's ratio 0.35), while the surrounding matrix of PS had an elastic modulus of 1.97 ± 0.2 GPa (Poisson's ratio 0.34) in Fig. 5m. These measurement results agree with the nominal values of the measured sample.

**Spatial resolution of the BMR NM.** To validate the spatial resolution of the proposed method, a triblock copolymer polystyrene-block-poly(ethylene/butylene)-block-polystyrene (SEBS)

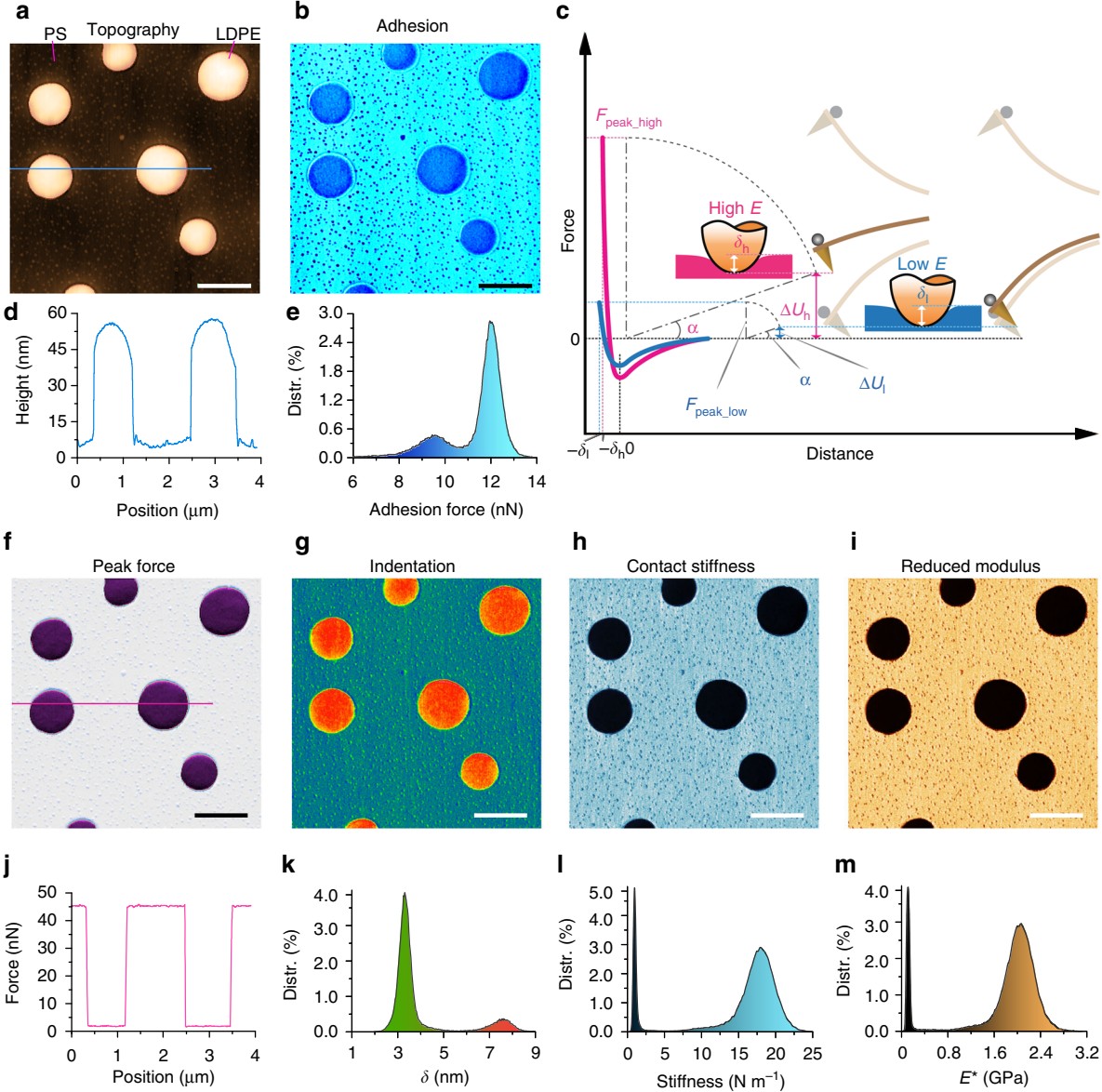

**Fig. 5** Nanomechanical maps of a polymer blend using Probe I. Top-right inset **c**: Scheme of the variable peak force control for indentation depth regulation on a heterogeneous surface. **a** Topography. **b** Map of the adhesion force. **d** Cross-section along the line shown in **a**, showing that the LDPE islands were up to 45–50 nm higher than the PS matrix. **e** Histogram of the adhesion force map described in **b**. **f** Map of the peak force, which was instantaneously regulated according to the contact stiffness on the PS and LDPE regions. **g** Map of the indentation depth δ. **h** Map of the stiffness. **i** Map of the elastic modulus. **j** Cross-section along the line shown in **f**. **k** Histogram of the δ map described in **g**. **l** Histogram of the stiffness map described in **h**. **m** Histogram of the elastic modulus map described in **i**. Scale bar, 1 μm

thin film was mapped with the probe I. As shown in Fig. 6a, the triblock copolymer is formed with ordered structures, and the cross section profile (Fig. 6c) indicates that the average height of the cylindrical patterns was approximately 11 nm. The patterns can be clearly seen in the adhesion map in Fig. 6b with a difference of 1 nN (Fig. 6d). The indentation and elastic modulus maps (Fig. 6e, f) demonstrate that the higher domains (stiffer domains) corresponded to the polystyrene (PS). The profiles (Fig. 6g, h) obtained from the indentation and elastic modulus maps exhibited a lateral resolution of ~30 nm that could be further improved by using a sharp probe with a small tip radius. The value of the elastic modulus on the PS domains was underestimated because of the soft ethylene/butylene components underneath.

Considering the broad modulus range, height, width and roughness distribution of the biological samples, *E. coli* TOP10

bacterial cells (freshly dried within 1 h) were scanned to test our method's adaptability and resolution. The results, as can be seen in Fig. 7 reveal that the BMR NM can be successfully applied with high resolution on a blend of soft and hard, continuous and discontinuous, linear and non-linear surfaces. As shown in Fig. 7a, the *E. coli* cell has several filamentous appendages, possibly flagella, with a diameter varying from 3 to 10 nm (height profile in Fig. 7d), and the average width of flagella measured to be 29.37 ± 14.73 nm (see Supplementary Note 5) with max and min as 76.7 and 13.8 nm, respectively. The cell structure can be clearly seen in the adhesion force map in Fig. 7b with a difference of 23 nN. The elastic modulus map (Fig. 7c) demonstrate that the bacterial flagella was clearly recognized from the substrate that is covered with proteins, lipids, salt and other matter. The cross-section profile (bottom of Fig. 7d) also proved a similar lateral resolution on the sample with discontinuous structures. The

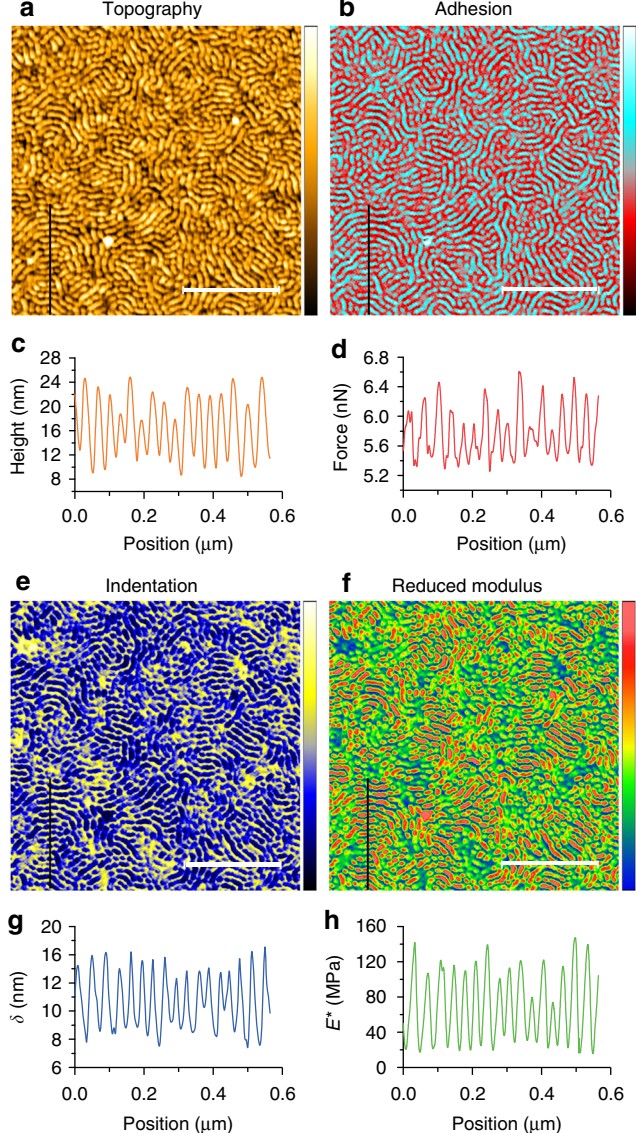

**Fig. 6** High spatial resolution NM of a SEBS polymer. The same probe was used to map a SEBS polymer. **a** Topography. **b** Map of the adhesion force. **c** Height profile obtained from **a**. **d** Profile of the adhesion forces obtained from **b**. **e** Map of the indentation depth ($\delta$). **f** Map of the reduced elastic modulus calculated from DMT theory. **g** Profile of $\delta$ obtained from **e**. **h** Profiles of the elastic modulus obtained from **f**. Scale bar, 500 nm

reduced elastic modulus of the flagella is ranging from about 400 MPa to 1.5 GPa that is strongly affected by the substrate due to its small diameter.

## Discussion

We developed a magnetic-drive peak force modulation atomic force microscope that has three characteristics. First, unlike the conventional piezo-drive force spectroscopy, this method provides direct bending actuation of the probe beam at off-resonance frequencies. Second, this method extremely broadens the nanomechanical measurement range of a single probe because it can directly measures the force and the indentation depth during acquisition of force-distance curves. Finally, the precise non-linearity compensation of the optical lever and the fully considered probe damping and inertial effects ensure accurate

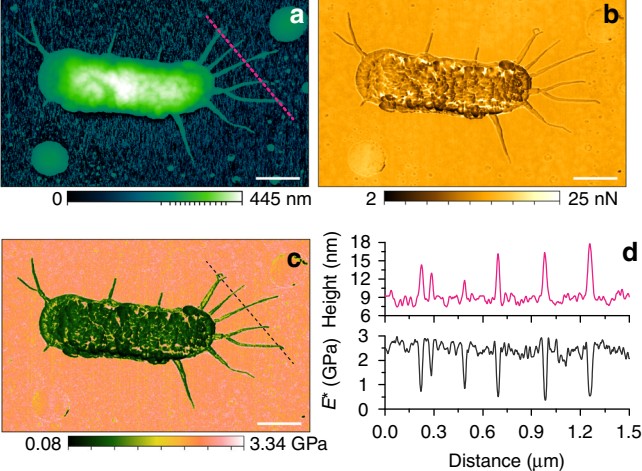

**Fig. 7** High spatial resolution NM of an *E. coli* TOP10 cell. **a** Topography. Maps of **b**, the adhesion force and **c**, the reduced elastic modulus calculated from DMT theory. **d** Profiles of the height (top) and reduced elastic modulus (bottom) obtained from **a**–**c**, respectively. Scale bar, 500 nm

measurements of the nanomechanical properties. With this approach, one can successfully map nanomechanical properties of different samples with elastic moduli ranging over four orders of magnitude using a single probe with high-resolution, both in liquid and air. In addition, with the further development of ultra-soft probes, it can be possible to access mechanical resolution below 1 kPa for NM of ultra-soft samples such as neurons. This method opens up the possibility to study mechanoresponse to stimuli, mechanical dynamics/real-time monitoring, and mechanical characterization of heterogeneous blends with wide elastic modulus variations, all of which require broad dynamic range of the probe. Furthermore, it reduces the error in results by eliminating the disturbing forces caused by indirect cantilever excitation methods as well as by removing the need to change the cantilever mid-experiment in liquid and air.

## Methods

**Theoretical model.** During the intermittent indentation, two external forces are applied to the tip: the equivalent magnetic drive force ($F_{drive}$) and the tip-sample interaction force ($F_{ts}$). Considering the hydrodynamic and the inertial forces of the probe[40], the tip displacement $z(x_{tip}, t)$ is given by the equivalent point-mass model:

$$m\ddot{z} + \frac{\omega_0 m}{Q}\dot{z} + kz = F_{drive} + F_{ts} \qquad (2)$$

where $m = k/\omega_0^2$ is the cantilever-tip effective mass and $\omega_0$, $Q$ and $k$ are the first resonance frequency, quality factor related to the hydrodynamic force and the spring constant of the probe, respectively. In air, $\omega_o$ and $Q$ are calibrated while the cantilever vibrates far from the sample (e.g., 1 μm) in the absence of capillary forces. In liquid, considering the distance-dependent hydrodynamic forces, $\omega_o$ and $Q$ are calibrated as the probe is very close to the substrate (e.g. with a distance >5 nm where the van der Waals forces can be ignored[41]). In both cases, the tip-sample interaction forces can be neglected, then eq. 3 becomes:

$$m\ddot{z}_{free} + \frac{\omega_0 m}{Q}\dot{z}_{free} + kz_{free} = F_{drive} \qquad (3)$$

From eqs. 2 and 3, the tip–sample interaction force $F_{ts}$ is given by

$$F_{ts} = m(\ddot{z} - \ddot{z}_{free}) + \frac{\omega_0 m}{Q}(\dot{z} - \dot{z}_{free}) + k(z - z_{free}) \qquad (4)$$

A key issue of using eq. 4 is transforming the photodiode voltage output $U(t)$ into the tip position $z(t)$. Since the force and torque lead to different vibration shapes, $U(t)$ is divided into $U_F(t)$ and $U_M(t)$ to determine the ($F_{ts}$) and the magnetic torque, respectively. We note that $U(t)$ is proportional to the deflection slope of the cantilever $\partial z(x_{laser}, t)/\partial x$ with a coefficient $\beta_{slope}$ that depends on the laser spot position. Thus, two deflection sensitivity factors, $\gamma_F$ and $\gamma_M$, of the optical lever are introduced for the force-induced deflection ($z_F$) and magnetic torque-induced deflection

($z_M$), respectively. The relationship between $U(t)$, $z(t)$ and the deflection slope of the cantilever at the laser spot can be expressed as

$$U(t) = U_F(t) + U_M(t)$$
$$= \beta_{\text{slope}} \left.\frac{\partial z_F(x,t)}{\partial x}\right|_{x=x_{\text{laser}}} + \beta_{\text{slope}} \left.\frac{\partial z_M(x,t)}{\partial x}\right|_{x=x_{\text{laser}}} \quad (5)$$
$$= \gamma_F z_F(x_{\text{tip}}, t) + \gamma_M z_M(x_{\text{tip}}, t)$$

$U_M(t)$ is generated by a stable magnetic drive and independent of the tip–sample interactions. It can be recorded at the very beginning of the measurement.

Since $\gamma_F \neq \gamma_M$ in most cases, the tip position cannot be directly measured using $U(t)$. Fortunately, $\gamma_F = \gamma_M$ (see Supplementary Note 2) if the laser spot is located at 2/3 (see Supplementary Note 6) of the cantilever length ($L_{\text{probe}}$) from its base. Thus, eq. 5 can be simplified as

$$U(t) = U_F(t) + U_M(t) = \gamma z(x_{\text{tip}}, t) \quad (6)$$

Then

$$z_{\text{free}}(x_{\text{tip}}, t) = z_M(x_{\text{tip}}, t) = U_M(t)\gamma^{-1} \quad (7)$$

$$z(x_{\text{tip}}, t) = U(t)\gamma^{-1} \quad (8)$$

Equation 6 can be used to extract the tip position directly from the photodiode signal, avoiding the conversion errors that may occur using the conventional force-distance-based methods, as seen in Fig. 1c. From eqs. 4, 8 and 7, the tip–sample interaction force is given by

$$F_{\text{ts}} = \left[ m(\ddot{U} - \ddot{U}_M) + \frac{\omega_0 m}{Q}(\dot{U} - \dot{U}_M) + k(U - U_M) \right]\gamma^{-1} \quad (9)$$

Once $k$, $\gamma$, $\omega_0$, $U_M(t)$ and $U(t)$ are known; the tip position and tip-sample interaction force can be determined from eqs. 8 and 9, respectively. As it is able to instantaneously detect the tip position and tip–sample interaction force, the FD curve can be easily constructed to obtain the elastic modulus by using the Derjaguin–Muller–Toporov (DMT) model

$$F_{\text{ts}} = \frac{4}{3} E^* \sqrt{R \delta^3} + F_{\text{adh}} \quad (10)$$

where $E^*$ is the reduced elastic modulus, $\delta$ is the indentation depth, $R$ is the tip radius, and $F_{\text{adh}}$ is the tip–sample adhesion force. Then, $\delta$ is recovered from eq. 8, and the zero-indentation position will occur at the pull-off point of the DMT model.

**Peak force regulation**. To map the nanomechanical properties of a heterogeneous surface, which is composed of materials with large elastic modulus variations, the main feedback of the maximum indentation force should be regulated at different regions to obtain effective indentation depths to accurately fit the contact mechanics. As shown in Fig. 5c, the peak force was confined to a relatively small value ($F_{\text{peak\_low}}$) with indentation depth ($\delta_l$) when the probe was located on a component with a low elastic modulus, while a relatively high value ($F_{\text{peak\_high}}$) was applied to obtain a similar indentation depth ($\delta_h$) when the tip was on a stiffer component. The regulated peak force ($F_{\text{peak}}$) was calculated using the following equation:

$$F_{\text{peak}} = \frac{4}{3} E_{\text{ref}} \sqrt{R \delta^3} - F_{\text{adh}} = \frac{4}{3}\left( \frac{N^{\lceil \log_N^{E^*} \rceil} + N^{\lfloor \log_N^{E^*} \rfloor}}{2} \right)\sqrt{R \delta_t^3} - F_{\text{adh}} \quad (11)$$

where $E^*$ is the instantly measured reduced elastic modulus, $N$ is an integer that is used to define the interval size for peak force regulation, $E_{\text{ref}}$ is a reference reduced elastic modulus located in the middle of the defined interval, and $\delta_t$ is the required indentation depth. In practice, when the peak force frequency is less than one-tenth of the first flexural resonant frequency, $F_{\text{peak}}$ can be directly controlled through the regulation of $U_F$[40] by $U_F = \Delta U = F_{\text{peak}} \tan \alpha = F_{\text{peak}}\gamma/k$.

**AFM probe calibration**. Two probes, B-lever HQ:NSC36/No Al (Probe I) and B-lever BL-RC150 VB (Probe II) of Olympus Biolever, were selected for the NM in air and liquid, respectively. The probe calibration includes determining the parameters of the microscope, such as the probe spring constant, force sensitivity and radius of the tip apex. The spring constants of the selected probes were determined to be 2.09 N m⁻¹ and 0.0058 N m⁻¹ using the mass loading method[42]. The resonant Frequency and Q-factor of the Probe I was calibrated as 73.85 kHz and 210, while 1.72 kHz and 1 for Probe II (in liquid), respectively. Rather than using the traditional force-induced probe-bending AFM calibration method, a nondestructive method with a flexure-hinge lever was used to calibrate the full range sensitivities of the optical lever[33]. Moreover, the nonlinearities of the force measurements were accurately compensated, and the linear range (with a deviation <5% of the full range) was extended to over 90% of the full range of the force measurement. The nonlinear force calibration substantially extended the indentation capability of the

soft probe for measuring the Young's Modulus of stiff materials. Thereafter, the tip radius of the first probe was calibrated as 10.4 nm by the reproducing Young's modulus of a standard polystyrene sample (2.7 GPa, Bruker Nano Inc.), and the tip radius of the second probe was calibrated as 36.3 nm by reproducing Young's modulus of a standard polydimethylsiloxane (PDMS) sample (2.5 MPa, Bruker Nano Inc.) in liquid.

**Samples preparation**. Seven samples were used in the experiments, four of which were standard calibration samples acquired from Bruker AFM probes, and the others were homemade samples. The standard samples include a polymer blend made of PS regions ($E_{\text{PS}} \approx 2.0$ GPa) and polyolefin elastomer (ethylene-octene copolymer) (LDPE) ($E_{\text{LDPE}} \approx 0.1$ GPa), a highly ordered pyrolytic graphite (HOPG) ($E_{\text{HOPG}} \approx 18$ GPa), and two types of PDMS samples with elastic moduli of 2.5 and 3.5 MPa. The use of standard calibration samples was intentional for the reasons of comparison. To enable the reader to compare their own or available-online results with ours. The SEBS polymer film with a thickness over 100 μm was prepared as described by Wang et al.[43]. The polyacrylamide gel (PA Gel) was made from a mix (3000 μL) of acrylamide (5% w/v, Aladdin CAS no. 79061), bisacrylamide (0.2% w/v, Aladdin CAS no. 110269), 4.5 μL of TEMED (Aladdin CAS no. 110189), 15 μL of ammonium persulfate (10% w/v, Aladdin CAS no. 7727540) and an appropriate amount of H₂O. The PA Gel sample for NM was prepared, as described by Janmey et al.[44]. The bacteria were cultured overnight in LB-medium at 37 °C in a shaker flask. An aliquot of the culture was pelleted by centrifugation, washed in deionized water and resuspended in Tris–HCl buffer. A 2 μl droplet of the bacterial suspension was drop casted onto a cleaned silicon or glass surface. The surfaces were rinsed twice with deionized water to remove non-adherent bacteria and allowed to dry. For the liquid NM of the Finegoldia Magna, the dried sample (with the substrate) was placed in a Petri dish and added 1 ml deionized water for scanning.

**Data availability**. The data that support the findings of this study are available from the corresponding author on reasonable request.

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

## Acknowledgements

We are grateful to Mingdong Dong, Aslan Husnu (Aarhus University, Denmark) for discussion. We thank You Wang, Shaoqing Liu, Qiang He, Ying Hu, Lei Wang, Chengliang Zhen, Meilin Chen, Ruiwen Wang, Wenjie Ge (Harbin Institute of Technology, China) for samples preparation. We acknowledge help from Chenyan Xu, Shaoyuan Sun (Research Center for Analysis and Measurement), Yongda Yan, Yang He from Harbin Institute of Technology for Facilities. This work was supported by the National Natural Science Foundation of China under Grants 61573121 and 51521003, and the National Key Research and Development Programme of China under Grant 2017YFA0207201.

## Author contributions

H.X., X.M. and L.S. conceived the method. H.X., X.M. and J.S. designed the system. H.Z. prepared the magnetic bead probe. X.M. and J.S. analysed the probe dynamics and built the theoretical model. H.Z. and X.F. calibrated the magnetic strength and the probe, and prepared the testing samples. X.M. and H.X. co-analysed the experimental and calculated data. All authors wrote the paper.

## Additional information

**Competing interests:** The authors declare no competing financial interests.

**Change history:** A correction to this article has been published and is linked from the HTML version of this paper.

