## [Peer Review File · Nature Communications]

Reviewers' Comments:

Reviewer #1:

Remarks to the Author:

This Ms. reports a new excitation method to drive a force microscope probe. By attaching a magnetic bead at the back of a microcantilever, the authors demonstrate that the probe can be approached and retracted from a sample by applying a varying magnetic force. The authors develop the concept and the instrument. The Ms. has also another relevant point, the instrument is applied to measure some nanomechanical properties, in particular, the Young's modulus. The Ms. reports the capability to measure the Young's modulus of very soft materials (1 kPa) and that of stiff surfaces (20 GPa). The authors emphasize that the broad nanomechanical spectroscopy range achieved because the magnetic drive enabled to control the application of the applied force from very small (35 pN) to large forces (300 nN).

Several magnetic excitation schemes have been implemented for AFM (W. Han, S. M. Lindsay, and T. Jing, Appl. Phys. Lett. 69, 41111996). However, the current proposal seems very robust, sensitive and easy to implement.

The Ms. is very well written. The figures are very informative and the data well reported. The magnetic drive method represents an original, sophisticated and simple method for exciting microcantilevers in water and air. The authors present convincing data about its nanomechanical capabilities by mapping the Young's modulus of polymer blends, hydrogels, HOPG and PDMS surfaces.

Overall, the Ms. has the quality, broad interest and relevance to be published in NCOMMS.

There are some comments.

1 The authors mention that the 230 nN could be generated by passing a 16.5 mA current through the solenoid. The data should be completed by providing the amplitude of the tip displacement in nm. The authors should also report the values of the amplitude in nm, as well as, the current passing through the solenoid to reach a force of 35 pN.

2 Some of the references mentioned in the Ms. describe experiments with optical tweezers (11, 14). Those experiments also measure the unbinding forces on some biomolecular systems, however, those experiments did not use an AFM. Of course, those references could be included in the Ms.. The authors need to place them in their context.

3 To calculate the tip-sample force the authors use an approximation (page 18) that considers the Newton equations for two cases, a tip interacting with a surface and non-interacting tip. Then, those equations are subtracted and the tip-sample force is expressed in terms of the free and interacting deflection. Could the authors provide an estimation of the applicability range of this approximation ?

Reviewer #2:

Remarks to the Author:

I read with interest this manuscript, which seems to be executed in a technically sound way. More uncertain is the significance of the work in a broader perspective: currently, from the evidence reported by the authors, it is highly unclear both the advantages and the performance over commercial benchmarks of similar nanomechanical spectroscopy methods. In my view, as it stands, this manuscript is technically correct, but does not have the broad relevance expected for publications in Nature Communications. A more technical platform (Appl. Phys. Lett or similar) would appear much more appropriate.

Major Comments:

1. What is the advantage of the described technique compared to already existing and commercially available techniques, e.g. Bruker PF-QNM?

2. The authors analysed almost only commercial samples (four samples from six were standard samples from Bruker). Additional biological samples with different mechanical properties must be analysed to prove the advantage, if any, of using the described technique.
3. How easy will be the preparation of the cantilever with a glued bead on top of it?
4. Lateral resolution is tested on continuous films only (Block copolymers). What would be the resolution on more challenging samples, such as filamentous objects? The Bruker PF-QNM method has been, for example, successfully tested on filamentous colloids such as amyloid fibrils back in 2011. Such a benchmark is a must on this case as well.

Minor comments: The word Topography is misspelled in all figures (Topgraphy)

Reviewer #3:

Remarks to the Author:

Broad modulus nanomechanical spectroscopy by magnetic-drive soft probes describes an AFM method for measurements based on the idea of using a AC magnetic field to drive the oscillations of a magnetically modified AFM cantilever. The paper is technically very sound as far as I can tell (I am no expert though on the mathematics outlined here) and the application examples shown are interesting. Being able to do measurements across a heterogeneous surface (in terms of stiffness) without changing the cantilever would certainly be useful.

The paper, however, suffers from two very significant flaws and until these have been addressed, the paper cannot be published in its current format.

Firstly, there is no mention or recognition of a very similar technique that has been around for nearly 20 years and was originally introduced by a company called Molecular Imaging (Arizona) under the trademark MAC-Mode. The company was since acquired by Agilent and is now Keysight Technologies. I recognise these instruments are not very well known and the companies involved have never fully disclosed all the details but in essence, MAC mode uses almost exactly the same type of solenoid as described in the current paper. The difference is in the cantilever design, the propriety MAC Mode AFM probes have a magnetic coating on top whereas the current paper uses a magnetic bead that presumably is glued on top.

The basics of the method was described in
Appl. Phys. Lett. 1996, 69, 4111.

See also Ultramicroscopy, 2007, 107, 299.

For some literature on MAC mode see:

Good technical description:

<https://www.agilent.com/cs/library/datasheets/Public/5989-5912EN.pdf>

Additional examples

<http://afmuniversity.em.keysight.com/PDFs/appnotes/5991-3672EN%20>

<MacMode%20Imaging%20of%20Biological%20Molecules%20with%20the%207500AFM.pdf>

<https://www.agilent.com/cs/library/applications/5989-6609EN.pdf> < BR>

<https://books.google.com.au/books?id=eUunjCXRYpC&pg=PA476&lpg=PA>

<476&dq=mac+mode+molecular+imaging&source=bl&ots=-YDQn4H2AU&sig=F2>

taCkAIALUuCkXvNF_iOOGbT58&hl=en&sa=X&ved=0ahUKEwj9_ZGMmblVAhWEq5Q

<KHR4ADsgQ6AEIPDAE#v=onepage&q=mac%20mode%20molecular%20imaging&f=false>

Pastushenko and Hinterdorfer at JPK University of Linz even developed a molecular recognition

version of MAC mode and published several papers on using this technology; see c.f. Ultramicroscopy 2000, 82, 227.
[https://doi.org/10.1016/S0304-3991\(99\)00146-1](https://doi.org/10.1016/S0304-3991(99)00146-1)

If the current paper was to be published the authors need to carefully explain the similarities and differences in the capabilities of the technology they described here vs the known MAC mode approach as well as adequately recognise the prior existence of MAC mode technology on which the current work appears to be based on or strongly related to.

The second issue is with regards to comparison. While not many people have access to MAC mode instruments and asking for comparison in imaging quality with MAC mode might be unfair, the authors ought at least to show some examples of how the sample they used appear when analysed with the now state-of-the-art Peak Force QNM (Bruker/Veeco). For instance, the granular surface apparent in Fig 3A – would that also show up with carefully adjusted settings in Peak Force QNM? Or for that matter (if access allows) in MAC mode AFM?

1. Response to comments from Reviewer #1

Dear Reviewer,

Please see our statements to your valuable comments on our manuscript:

I. General Comments:

This Ms. reports a new excitation method to drive a force microscope probe. By attaching a magnetic bead at the back of a microcantilever, the authors demonstrate that the probe can be approached and retracted from a sample by applying a varying magnetic force. The authors develop the concept and the instrument. The Ms. has also another relevant point, the instrument is applied to measure some nanomechanical properties, in particular, the Young's modulus. The Ms. reports the capability to measure the Young's modulus of very soft materials (1 kPa) and that of stiff surfaces (20 GPa). The authors emphasize that the broad nanomechanical spectroscopy range achieved because the magnetic drive enabled to control the application of the applied force from very small (35 pN) to large forces (300 nN).

Several magnetic excitation schemes have been implemented for AFM (W. Han, S. M. Lindsay, and T. Jing, *Appl. Phys. Lett.* 69, 41111996). However, the current proposal seems very robust, sensitive and easy to implement.

The Ms. is very well written. The figures are very informative and the data well reported. The magnetic drive method represents an original, sophisticated and simple method for exciting microcantilevers in water and air. The authors present convincing data about its nanomechanical capabilities by mapping the Young's modulus of polymer blends, hydrogels, HOPG and PDMS surfaces.

- **Reply:** Thanks to the reviewer for positive comments and kind reminding. The work (Han, Wenhai, S. M. Lindsay, and Tianwei Jing. A magnetically driven oscillating probe microscope for operation in liquids." *Applied Physics Letters* 69.26 (1996): 4111–4113.) has been cited (ref. 35) in our revised paper because of authors' significant contributions in the magnetic excitation.

To my knowledge, the above mentioned work is the groundbreaking results of the magnetically driven AFM in liquid condition. This method provides a direct and reliable actuation of the AFM probe over a broad frequency range with excitation amplitudes of a few nanometers, which facilitates high-resolution imaging of samples that are smaller than the tip radius by means of small asperities on the tip.

Inspired by the commercialized resonant magnetic drive method, we have developed this magnetic drive peak force modulation scanning method. In addition to the same way of driving, there are some major differences between these two methods:

- i) Unlike the current magnetically driven resonant mode AFM for tapping or non-contact imaging, our method is to drive the probe in a non-resonant mode for the force-distance curve based

nanomechanical mapping. The former requires a driving force of several nanonewtons or less, while the latter may need a driving force of several hundreds of nanonewtons to mapping a stiff sample surface.

- ii) Magnetic microbeads, instead of nano-magnetic coating, is used to get a greater driving force (with an optimized solenoid and its driver) that is proportional to the cube of diameter of the magnetic component.
- iii) In addition, instead of vibrating the probe holder or sample with the piezoactuator (currently widely used peakforce AFM), only cantilever beam is sinusoidally oscillated in this method by the magnetic torque at the selected frequencies. The vibration coupling noise is minimized owing to the negligible total mass of the cantilever beam and the attached magnetic bead ($\text{\O}3\text{--}15\ \mu\text{m}$) both in air and liquid environments. With this method, we can directly identify the indentation depth and force in real time from the cantilever response during the entire measurement process. As a result, the range of the modulus in nanomechanical mapping can be greatly broaden.

Moreover, as shown in Figure 1, the magnetic bead is in our method placed directly on top the tip by precision micromanipulation technique, unlike magnetic coatings or glued magnetic particle modifications. This results in the local magnetic excitation of the cantilever, directly aligned along the tip itself as can be seen in the below figure. This way the tip position can be directly controlled. In contrast to magnetic thin film coatings or random shape and position magnetic particles, the dynamics of the exciting force can be accurately followed and remain stable, yielding in constant k value ergo applied force. This becomes, especially important when it comes to the operation in liquid environment.

Figure 1: Configurations of different magnetic drive probes.

II. Major Comments:

- 1) The authors mention that the 230 nN could be generated by passing a 16.5 mA current through the solenoid. The data should be completed by providing the amplitude of the tip displacement in nm. The authors should also report the values of the amplitude in nm, as well as, the current passing through the solenoid to reach a force of 35 pN.

■ **Reply:** We thank the reviewer for the comments and understand the concerns. In figure 2a, we used a force modulation probe (B- lever of HQ:NSC36/No AI, nominal spring constant of $k = 2$ N/m) to estimate the equivalent magnetic driving force A_f that is calculated by:

$$A_f = kA_m \sqrt{\left[1 - \left(\frac{\omega}{\omega_o}\right)^2\right]^2 + \left(\frac{1}{Q} \frac{\omega}{\omega_o}\right)^2} \quad (1)$$

where A_m , ω_o and Q are the the oscillating amplitude with the off-resonance frequency of ω , the 1st resonant frequency and Quality factor of the probe, respectively. In the case of the measurement in air, this calculation is simplified as $A_f = kA_m$ when $\omega \ll \omega_o$ and $Q \gg 1$ (normally $Q > 100$ in air for the common AFM probes).

In the actual use, probes with different spring constants (e.g. 5 N/m) can be selected to calibrate magnetic driving force. In order to intuitively read the value of the driving force, we convert the probe vibration amplitude into force. Therefore, it is better to keep the expression of the driving force in the force unit rather than the deflection (nm) of the probe cantilever.

In addition, the magnitude of the magnetic driving force is proportional to the cube of the diameter of the magnetic bead. The magnetic force of 230 nN is generated by the magnetic bead with a size of $\varnothing 11.4 \mu\text{m}$, while a magnetic bead with a size of $\varnothing 3.8 \mu\text{m}$ is used to drive the 0.006 N/m probe in liquid. Considering the liquid damping ($Q \approx 1$ for this probe) and the inertial effects, the magnetic driving force (250 Hz) is calibrated from the above equation, as shown in Figure 2. The p-p current of ~ 5.3 mA is applied to the solenoid to vibrate the probe with an amplitude of 200 nm (equivalent force: 1.2 nN), while ~ 0.17 mA for the vibrating amplitude of ~ 5.8 nm to produce a equivalent driving force of 35 pN. We would like explain to the reviewer that the “35 pN” in the manuscript presents the contact (indentation) force, rather than the magnetic driving force. This value, related to the indentation depth, has no direct relationship with the driving current. We provide in detail explanations in the Supplementary Information and a change also has been made in the manuscript in order to prevent possible confusion.

- 2) Some of the references mentioned in the Ms. describe experiments with optical tweezers (11, 14). Those experiments also measure the unbinding forces on some biomolecular systems, however, those experiments did not use an AFM. Of course, those references could be included in the Ms.. The authors need to place them in their context.

■ **Reply:** We appreciate the reviewer’s attention to the references, those two are certainly misplaced. In the previous versions of our manuscript, all Force Spectroscopy methods, including Optical and Magnetic Tweezers, were discussed in the introduction. As we reshaped the manuscript to its final form, we decided to exclude these parts since we were convinced by the staggering amounts of publications that nanomechanical applications and its market are dominated by AFM-based methods. During this transformation, more attention should had paid. We replaced the Ref. 11 with Hugel, T. and Seitz, M., The Study of Molecular Interactions by AFM Force Spectroscopy. *Macromol. Rapid Commun.*, 22, 2001, 989–1016 and Ref. 14 with S. Izrailev, S. Stepaniants, M.

Figure 2: Calibrated magnetic drive force (A_f) when the probe (with a $\text{\O}3.8 \mu\text{m}$ magnetic bead) was oscillated at 250 Hz with different driving currents (A_c) in water.

Balsera, Y. Oono, K. Schulten, Molecular dynamics study of unbinding of the avidin-biotin complex, *Biophysical Journal*, Volume 72, Issue 4, 1997, 1568–1581.

- 3) To calculate the tip-sample force the authors use an approximation (page 18) that considers the Newton equations for two cases, a tip interacting with a surface and non-interacting tip. Then, those equations are subtracted and the tip-sample force is expressed in terms of the free and interacting deflection. Could the authors provide an estimation of the applicability range of this approximation?

■ **Reply:** We thank the reviewer for the comment and this is also the main issue in the modelling of the probe dynamics, whether for our method, or Bruker Peakforce QNM approach. We would like to discuss this issue in the nanomechanical mapping in air and the liquid environment separately:

(i) For the application in air, the surrounding damping of the probe is quite low. When $\omega \ll \omega_o$ and $Q \gg 1$, e.g. $\omega = 1,2 \text{ KHz}$, $\omega_o = 73.85 \text{ KHz}$, $Q = 210$ for one of the probes we used (2 N/m), from the eq.2 in the main context, the damping and inertial force effected can be ignored. The similar conclusion can be also found in ref. 39 (ACS Nano 2016, 10, 7117–7124.) of the manuscript, which confines the max. ω at $\sim 2\%$ of ω_o . Therefore, if the above conditions are met, the measurement of the magnetic driving force and the quality factor in the air environment is not affected by the probe-substrate distance.

(ii) In the case of the liquid environment, Q and ω_o are significantly reduced due to the large damping of the liquid. Especially when the probe close to the sample surface, the hydrodynamic force between the probe and the sample surface is increased considerably. On the other side, due to the need for rapid scanning, the driving-frequency ω of FD curve measurement often exceeds 2% of the first resonant frequency of the probe, or even up to 10–20%. Thus, the effects of damping and inertia forces can not be ignored in the liquid environment. The driving force and the quality factor Q are recommended to be calibrated with the probe being very close to but not touching the substrate (e.g. $>5 \text{ nm}$ where the van der Waals force cannot be observed^[1]).

Based on the above discussion, we realize that expression is not accurate enough. So we modify the sentence as: **In the air environment, ω_o and Q are calibrated when the cantilever vibrates far from the sample (e.g. $1 \mu\text{m}$) in the absence of capillary force, while in liquid, considering the distance-dependent hydrodynamic force, ω_o and Q are calibrated as the probe is very close to the substrate ($> 5 \text{ nm}$ where the van der Waals force can be ignored^[1]). In both cases, the tip-sample interaction forces can be neglected, eq 3 becomes:**

Reference

1. Stifter, T., Marti, O., & Bhushan, B. (2000). Theoretical investigation of the distance dependence of capillary and van der Waals forces in scanning force microscopy. *Physical review B*, 62(20), 13667.

2. Response to comments from Reviewer #2

Dear Reviewer,

Please see our statements to your valuable comments on our manuscript:

I. General Comments:

I read with interest this manuscript, which seems to be executed in a technically sound way. More uncertain is the significance of the work in a broader perspective: currently, from the evidence reported by the authors, it is highly unclear both the advantages and the performance over commercial benchmarks of similar nanomechanical spectroscopy methods. In my view, as it stands, this manuscript is technically correct, but does not have the broad relevance expected for publications in Nature Communications. A more technical platform (Appl. Phys. Lett or similar) would appear much more appropriate.

■ **Reply:** We thank the reviewer for the comments and understand the concerns. We try to address the comparative advantages and performance of our work in dedicated sections below. Here, the significance of the work in a broader perspective will be discussed.

According to Bruker: "Utilizing five separate patents, PeakForce Tapping has led to over 1,000 peer-reviewed publications in the first five years since its release, generating nearly 3,000 citations. This adoption rate has surpassed even that of TappingMode."^[1] Considering how TappingMode spread like a wild fire in biological and material community after its introduction, the PF QNM's achievement is truly outstanding. Moreover, Bruker's PF QNM is only one of the Nanomechanical Mapping (NM) methods, the overall impact of NM in academia and industry is undoubtedly tremendous. It is not surprising that PF QNM alone has surpassed TappingMode because just imaging alone is not enough to identify the differences between various matters. Although, in biology structure and function is closely related, matter with similar structure can have different mechanical properties ergo function. For example, Howard *et al.* shows how Tumour exomes, though indistinguishable by shape, display differential mechanical and complement activation properties dependent on malignant state^[2]. Beyond medicine and biology such experiments have a broad application range including but not limited to dairy and food industry to distinguish emulsions, precipitations, fibers etc., in pharmaceutical industry to distinguish drugs, drug delivery systems etc., in energy industry to distinguish thin film coatings, multi-layer compositions, battery components etc., in automotive industry for the analysis of defects, all that cannot be distinguished solely based on shape. It is, therefore, clear that NM will be more in demand over the years. However, one can also see that in order to distinguish a wide range of mixed samples (and for other reasons discussed in detail below), a wide range of moduli detection is required. Yet, currently available systems can only provide discrete NM solutions for quite narrow modulus spectra. In this study, all discrete modulus detection ranges are unified into two spectra. Each of which covers the most-in-demand applications or sample types.

This work considerably expands the applications of NM, and clears the path of its further development. Thus, its communication to broad audiences is of utmost importance.

II. Major Comments:

- 1) What is the advantage of the described technique compared to already existing and commercially available techniques, e.g. Bruker PF-QNM?

- **Reply:** Nanomechanical Mapping (NM) relies on understanding, utilizing and controlling the cantilever properties to achieve quantitative results. Due to the limitations in calculating/calibrating the non-linear cantilever dynamics and its integration with the hard- and soft-ware of the instrument, current systems in the market can only provide discrete NM solutions. That is, each cantilever can work within a narrow range of elastic moduli, creating a demand for cantilever change and calibration for samples with moduli range larger than that of the cantilever. This issue is so drastic that even for a moduli range from 2.5 MPa to 9 MPa of single proteins could not have mapped with a single cantilever.^[3] The requirement to change and recalibrate the cantilever(s) is time and money consuming, perhaps more importantly it restricts the NM for:

i) Studying mechanoresponse to stimuli

Matter can change mechanical properties given certain stimuli such as light, heat, pH, electricity etc. These changes can be extreme, from very soft gels to very hard solids, for example UV curable glues. Due to very narrow moduli range of current NM methods, currently, it is not possible to study matter's mechanoresponse to stimuli.

ii) Mechanical dynamics/real-time monitoring

Matter's mechanical properties can change given time and/or use. In order to investigate the aging and wear effects real-time studies should be conducted. However, if the mechanical dynamics scale over the range of the cantilever's range, such experiments cannot be executed. Thus, these studies require broad modulus ranges.

iii) Mechanical identification

Mixed matter cannot always be identified by their shape, weight etc. They may be similar in those aspects, rendering them indistinguishable. NM can help the identification of mixed particles based on their mechanical properties i.e. same group of particles will share same mechanical properties. However, these mixed matter can often have a wide range of modulus. Preventing or limiting their mechanical identification with currently available systems.

iv) Error reduction

The mechanical values of the sample are extracted by fitting a desired contact mechanics model to the force distance curve. These models such as Hertz, Sneddon, Derjaguin-Muller-Toporov, Johnson-Kendall-Roberts, Oliver-Pharr, etc. are based on the tip shape geometry, and the cantilever's Young's modulus. By changing the cantilever, these parameters are altered hence a superfluous uncertainty is introduced.

Moreover, PF-QNM and similar modes use piezo excitation to oscillate the cantilever, which causes a phenomenon known as forest of peaks and hinders the calibration and the operation of the instrument. Shaking the whole cantilever, cantilever chip, chip holder and its spring (or other locking mechanism) obviously alters the hydrodynamics which especially in fluid operations increases the error, can even disturb the sample. All these issues are irrelevant to direct cantilever excitation operations, such as ours.

The main advantage of our system compared to commercially available methods is the unification of multifarious modulus spectra into a single-split modulus spectrum as depicted in Fig. 3. This way we can address the above stated issues better than ever before.

In addition, in the section 2.3 (page 5) of the Bruker Peakforce QNM User Guide (http://www.torontomicrofluidics.ca/cms/manuals/peak_force.pdf) gives suggestions for probe selection: **"It is important to choose a probe that can cause enough deformation of the sample and still**

Figure 3: Elastic moduli of various common matter^[4-7] (in black), the working moduli spectra of cantilevers above the meter shows this work, below the meter is of current systems. * and ** are prototypical cantilevers.

retain high force sensitivity. Therefore cantilever stiffness should be selected based on the sample stiffness. Brukers recommendations are shown in Table 2.3a.” From this table, four probes (0.5, 5, 40 and 200 N/m) are recommended to cover the elastic modulus range 1 MPa to 20 GPa. In contrast, our method can cover this range with only one soft probe (2 N/m) because of three characteristics: (i) direct cantilever excitation with magnetic drive at off-resonance frequencies and sufficient indentation forces for broad nanomechanical measurement, (ii) direct force and indentation depth measurement during force-distance curve acquisition, (iii) precise nonlinearity compensation of the optical lever and the fully considered probe damping and inertial effects. With this approach, one can successfully map nanomechanical properties of different samples with elastic moduli ranging over four orders of magnitude using a single probe with high-resolution, both in liquid and air. In particular, to our knowledge, the ability to nanomechanically map heterogeneous surfaces with large elastic modulus variations is reported for the first time using the regulated peak force modulation AFM approach.

Last but not least, the work presented here, can be commercialized to a stand-alone product or can be adopted by currently available AFMs in the market. Potentially, BMR NM can be introduced to the market as a kit consisting of a coil embedded sample scanner-controller, a box of magnetic-bead decorated and calibrated cantilevers and the software. Surely, users can simply modify their own setups by adding a coil to their sample scanner, making and calibrating the cantilevers as mentioned in the manuscript. We encourage and support the DIY (Do It Yourself) trend, and welcome questions from those who seek assistance to build their own systems.

2) The authors analysed almost only commercial samples (four samples from six were standard samples from Bruker). Additional biological samples with different mechanical properties must be analysed to prove the advantage, if any, of using the described technique.

■ **Reply:** Thanks for the reviewer's comments. As the reviewer points out, we used the standard calibration samples from Bruker for PF QNM. This was indeed on purpose to show our results' comparability of resolution with the best representative of the current-state-of-the-art, on top of which we achieve moduli unification. We took up the reviewer's suggestion and conducted experiments on biological samples. We have chosen to work with three types of bacteria cells (*E. coli* TOP10, *E. coli* TSK and *Finogoldia Magna* bacteria) for such samples provide the body of the bacterium with altering elastic moduli. The bacteria were imaged in liquid (*Finogoldia Magna*) and air (*E. coli* TOP10 and TSK) by the proposed method, as shown in Figure 4. 6 and 7.

In Figure 4, the high-contrast maps of indentation, stiffness and reduced elastic modulus showed the soft bacteria has a clear difference from the hard glass substrate. The background noise of the indentation depth is greatly compressed at sub-nm which mainly due to the system noises, as well as softened or even floating coverings (proteins, lipids, DNA and other matter) on the glass substrate. Although the silicon probe is not capable of measuring the glass surface, as it is common sense, the clear and high-contrast elastic modulus map (from 40 MPa–25 GPa) verified the BMR NM capability. We have included our results and discussion of Figure 4 in the Supplementary Information.

Figure 4: **Nanomechanical mapping of *Finogoldia Magna* bacteria in liquid using Probe I.** (A) Topography. Maps of (B) the adhesion force, (C) the indentation depth and (D) the reduced elastic modulus. (a–d) Corresponding cross-section profiles obtained from A–D, respectively. Scale bar, 1 μm.

3) How easy will be the preparation of the cantilever with a glued bead on top of it?

■ **Reply:** In order to explain the ease of cantilever preparation, we have created the below section and added it in the Supplementary Information.

Requirements of magnetic microbead probe preparation are similar to colloidal probe methods. These are: i) Glue deposition on the cantilever, ii) Placement of the bead, iii) Calibration of the spring constant after the glue is dried. In order to achieve these steps with haste, a pneumatic micromanipulation system is developed as shown in SI Fig. 5(a). The micromanipulation system

mainly consists of a micro vacuum pump, a solenoid valve, a vacuum regulator and micropipettes (supported on a motorized stage with a motion resolution of 50 nm). Thanks to the top-view and side-view optical microscopes, glue (DP760 epoxy adhesive) and the microbead can be sequentially deposited and released on the target position with high precision, respectively. Fig. 5(b) shows micropipettes with different aperture diameters which are used to manipulate magnetic microbeads with a diameter range of 3–15 μm . Fig. 5(c) shows the optical microscopy images captured during the adhesive bonding process. One micropipette is utilized to draw an appropriate volume of glue with the action of capillary or suction pressure (if more volume of glue is needed) of about -5 kPa for 20 seconds. Glue is then deposited to the back side of the cantilever (unlike colloidal probe cantilevers) by applying an insufflation pressure of about 5 kPa for 1–2 seconds. The pressure force is larger than the friction drag at the interface of glue-micropipette wall. Another micropipette is used to pick up a microbead by applying a suction pressure to overcome the adhesion at the microbead-substrate interface. The microbead is then released at the target position on the adhesive droplet. Finally, the AFM probe is unloaded and placed in a vacuum oven for 12 hours at 60°C. After all, the magnetic bead is magnetized in a pulse magnetic field ($\sim 5\text{ T}$) in the direction of the longitudinal axis of the cantilever. The cantilever is then calibrated for broad modulus range operations in air and liquid.

Figure 5: (a) Schematic diagram of experimental setup includes a pneumatic control system for adhesive bonding of magnetic microbead on the AFM probe. (b) Micropipettes with different aperture diameter used to pick-and-place different size of microbeads. (c) Optical microscopy images captured during the process of preparation of the magnetic bead probe. Scale bar, 20 μm

Preparing modified cantilevers can take some time, which holds true for all type of modified probes i.e. physically, chemically or biologically. Unlike some of other modified probes, the ones used in this study can be easily commercialized and stored for a relatively long time. When considering the time consumption, however, one should account for the way we are using these probes, to unify the discrete moduli spectra. In other words, with a single probe we cover the moduli range of multiple probes therefore, the time is saved from replacing and re-calibrating each probe.

4) Lateral resolution is tested on continuous films only (Block copolymers). What would be the resolution on more challenging samples, such as filamentous objects? The Bruker PF-QNM method has been, for example, successfully tested on filamentous colloids such as amyloid fibrils back in 2011. Such a benchmark is a must on this case as well.

■ **Reply:** We are grateful for the reviewer's suggestion, and we do believe that scanning the sample with a discontinuous pattern can better reflect the system's resolution. At the beginning, we bought three different materials (insulin amyloid fibrils, α -synuclein amyloid fibrils and glucagon amyloid fibrils) and prepared the samples according to the protocols described in the articles^[8–10]. Unfortunately, since we are not specialized in chemistry or biology, we tried many times but failed. Fortunately, we found that the flagellum (fibrous protein chain) of bacteria is also with a diameter less than 10 nanometers, so we cultured and scanned two kinds of bacteria *E. coli* TOP 10 (freshly dried within 1 hour) and *E. coli* TSK (dried for more than 48 h) for the demonstration of the lateral resolution, the scan results (maps of Topography, Adhesion and Reduced elastic modulus) are shown in Figure 6 and 7. From the section-profiles (Figure 6d and Figure 7d) of the Topography images, it is found that the diameter of the flagella is in the range of 3–10 nm and the flagella structure can be clearly seen in the adhesion force and reduced elastic modulus maps.

To quantify the later resolution, as shown in Figure 8 (The grayscale is properly adjusted to the height of the flagella for clear display, which has no influence on the measurement results), statistical calculation is performed for lateral resolution with 40 measurements on different positions that are evenly distributed on the flagella, and the average width of flagella to be 29.37 ± 14.73 nm with max and min to be 76.7 and 13.8 nm, respectively. Similarly, as seen in Figure 9, the average width of flagella (from 20 measurements) to be 24.88 ± 4.5 nm with max and min to be 47.7 and 20.8 nm, respectively. The results, as can be seen in above figures reveal that the BMR NM can be successfully applied with high resolution on a blend of soft and hard, continuous and discontinuous, linear and non-linear surfaces.

Mapping results of the *E. coli* TOP 10 has been added to the manuscript. Results of the *E. coli* TSK and the later resolution calculation method have been added to the Supplementary Information.

Figure 6: High spatial resolution nanomechanical mapping of an *E. coli* TOP 10 cell. (a) Topography. Maps of (b) the adhesion force and (c) the reduced elastic modulus. (d) Profiles of the height (top) and reduced elastic modulus (bottom) obtained from a and c, respectively. Scale bar, 500 nm.

Figure 7: High spatial resolution nanomechanical mapping of an *E. coli* TSK cell. (a) Topography (nm). Maps of (b) the adhesion force (nN) and (c) the reduced elastic modulus (GPa). (d) Profiles of the height (top) and reduced elastic modulus (bottom) obtained from a and c, respectively. Scale bar, 500 nm.

Figure 8: Statistical calculation of lateral resolution with 40 measurements on different positions that are evenly distributed on the flagella.

Figure 9: Statistical calculation of lateral resolution with 20 measurements on different positions that are evenly distributed on the flagella.

III. Minor comments:

1) The word Topography is misspelled in all figures (Topgraphy).

■ **Reply:** We are grateful for the reviewer's notification. This mistake and some typos have been corrected.

Reference

(1) Bruker. Bruker PeakForce QNM Brochure, <http://mbns.bruker.com/acton/attachment/9063/f-025c/0/-/-/-/-/PeakForce%20Tapping%20-%20B080-RevA2.pdf> (2017).

- (2) Whitehead, B. et al. Tumour exosomes display differential mechanical and complement activation properties dependent on malignant state: implications in endothelial leakiness. *Journal of extracellular vesicles* 4, 29685 (2015).
- (3) Perrino, A. P. & Garcia, R. How soft is a single protein? The stress-strain curve of antibody pentamers with 5 pN and 50 pm resolutions. *Nanoscale* 8, 9151-9158 (2016).
- (4) Akhtar, R., Sherratt, M. J., Cruickshank, J. K. & Derby, B. Characterizing the elastic properties of tissues. *Materials Today* 14, 96-105 (2011).
- (5) Kaul, A., Gangwal, A. & Wadhwa, S. Nanoscale measurements for computing Young's modulus with atomic force microscope. *Current Science*, 1561-1566 (1999).
- (6) Gribova, V., Crouzier, T. & Picart, C. A material's point of view on recent developments of polymeric biomaterials: control of mechanical and biochemical properties. *Journal of materials chemistry* 21, 14354-14366 (2011).
- (7) University of Cambridge, D. o. E. Property information. (2002).
- (8) Zhou, X., Zhang, Y., Zhang, F., Pillai, S., Liu, J., Li, R. & Zhang, Y. (2013). Hierarchical ordering of amyloid fibrils on the mica surface. *Nanoscale*, 5(11), 4816-4822.
- (9) Sweers, K., Van Der Werf, K., Bennink, M., & Subramaniam, V. Nanomechanical properties of α -synuclein amyloid fibrils: a comparative study by nanoindentation, harmonic force microscopy, and Peakforce QNM. *Nanoscale research letters*, 6(1), 270, (2011).
- (10) Zhou, X., Cui, C., Zhang, J., Liu, J., & Liu, J. Nanomechanics of individual amyloid fibrils using atomic force microscopy. *Chinese Science Bulletin*, 55(16), 1608-1612, (2010).

3. Response to comments from Reviewer #3

Dear Reviewer,

Please see our statements to your valuable comments on our manuscript:

I. General Comments:

Broad modulus nanomechanical spectroscopy by magnetic-drive soft probes describes an AFM method for measurements based on the idea of using a AC magnetic field to drive the oscillations of a magnetically modified AFM cantilever. The paper is technically very sound as far as I can tell (I am no expert though on the mathematics outlined here) and the application examples shown are interesting. Being able to do measurements across a heterogeneous surface (in terms of stiffness) without changing the cantilever would certainly be useful.

■ **Reply:** We thank the reviewer for acknowledging the soundness of the technicality and usefulness of our unique instrumental capacity. Should the outlined mathematics be seem repulsive, it is simply because we wished to be thorough with the theory as well as the experiments.

Like the reviewer mentions, it is indeed of great importance to be able to employ a single cantilever for nanomechanical measurements, not only for the ease of use but also for scientific correctness. Unfortunately, today's state-of-the-art can only provide discrete solutions for nanomechanical mapping (NM) with the need for change of cantilever to cover a wide modulus range. This strays the experiment from true quantitative measurement. The most important issue is due to the fact that the change of cantilever changes the contact area, effective contact spring constant and indentation, which are the core of interaction force modelling and calculations for NM. The use of a single cantilever for the whole stiffness range of a sample, enables reliable calculations thus remarkably increases the accuracy and precision of the results.

II. Major Comments:

The paper, however, suffers from two very significant flaws and until these have been addressed, the paper cannot be published in its current format.

- 1) Firstly, there is no mention or recognition of a very similar technique that has been around for nearly 20 years and was originally introduced by a company called Molecular Imaging (Arizona) under the trademark MAC-Mode. The company was since acquired by Agilent and is now Keysight Technologies. I recognise these instruments are not very well known and the companies involved have never fully disclosed all the details but in essence, MAC mode uses almost exactly the same type of solenoid as described in the current paper. The difference is in the cantilever design, the propriety MAC Mode AFM probes have a magnetic coating on top whereas the current paper uses a magnetic bead that presumably is glued on top.

The basics of the method was described in Appl. Phys. Lett. 1996, 69, 4111.

See also Ultramicroscopy, 2007, 107, 299.

For some literature on MAC mode see:

Good technical description:

<https://www.agilent.com/cs/library/datasheets/Public/5989-5912EN.pdf>

Additional examples

<http://afmuniversity.em.keysight.com/PDFs/appnotes/5991-3672EN%20MacMode%20Imaging%20of%20Biological%20Molecules%20with%20the%207500AFM.pdf>

<https://www.agilent.com/cs/library/applications/5989-6609EN.pdf>

https://books.google.com.au/books?id=eUunjCXRYPC&pg=PA476&lpg=PA476&dq=mac+mode+molecular+imaging&source=bl&ots=-YDQn4H2AU&sig=F2taCkAIALUuCkXvNF_i0OGbT58&hl=en&sa=X&ved=0ahUKEwj9_ZGMmblVAhWEq5QKHR4ADsgQ6AEIPDAE#v=onepage&q=mac%20mode%20molecular%20imaging&f=false

Pastushenko and Hinterdorfer at JPK University of Linz even developed a molecular recognition version of MAC mode and published several papers on using this technology; see c.f. Ultramicroscopy 2000, 82, 227.

[https://doi.org/10.1016/S0304-3991\(99\)00146-1](https://doi.org/10.1016/S0304-3991(99)00146-1)

If the current paper was to be published the authors need to carefully explain the similarities and differences in the capabilities of the technology they described here vs the known MAC mode approach as well as adequately recognise the prior existence of MAC mode technology on which the current work appears to be based on or strongly related to.

- **Reply:** We appreciate the reviewer's healthy skepticism. Like the reviewer references, magnetically driven AFM probes and their applications have been reported in the literature for a longtime. After the introduction of Magnetic Force Microscopy (MFM) in 1987 by Martin and Wickramasinghe^[1] it was realized that the magnetic forces can be used directly to bend the cantilever. Especially after the success of other direct probe actuation methods such as photothermal heating in 1992 by Marti et al.^[2] many researchers have been using magnetic cantilevers either by coating them with magnetic materials such as cobalt via thin film deposition methods or gluing small magnets to the backside of the cantilever. Thus, their use was extended into AC imaging such as in the article^[3], the reviewer was kind to share with us. We would like to remind that MAC is an imaging-only method. From this point onward MAC method and the likes have been employed for direct probe control for imaging, until in 1998 Schemmel and Gaub introduced single molecule force spectrometer with magnetic force control and inductive detection^[4]. With this development magnetically coated or magnetic particle decorated cantilevers were driven for force spectroscopy either with the force of a magnetic field gradient or the torque of any perpendicular field. We would like to point out that all these methods were employed for conventional force spectroscopy in contrast to quantitative force mapping which is the most advanced version of force spectroscopy, then nanomechanical mapping which is the most advanced version of quantitative force mapping. In our study, we used magnetic torque driven cantilevers for the most advanced version of nanomechanical mapping. For this reason, as well as the length constraints of a communication article, we limited our discussion to nanomechanical mapping modes hence cited related articles.

Moreover, as shown in Figure 10, the magnetic bead in our method is placed directly on top the tip by precision micromanipulation technique, unlike magnetic coatings or glued magnetic particle modifications. This results in the local magnetic excitation of the cantilever, directly aligned along the tip itself as can be seen in the below figure. This way the tip position can be directly controlled. In contrast to magnetic thin film coatings or random shape and position magnetic particles, the dynamics of the exciting force can be accurately followed and remain stable, yielding in constant

k value ergo applied force. This becomes, especially important when it comes to the operation in liquid environment.

Figure 10: Configurations of different magnetic drive probes.

We would like to underline that in this paper, we are not focused on the instrumentation but rather our instrument's ability to provide a long sought after answer to the infamous discrete moduli range problem. That is owing to our novel approaches for cantilever calibration, parametric actuation, acquisition, modeling, calculation & visualization of the force-distance data, and mapping algorithms. In the Fig. 11, we humbly try to show each breakthrough in AFM-based force-distance data handling with oversimplification:

Figure 11: Breakthroughs in AFM-based force-distance data handling.

To summarize, the MAC mode rests within the first box (AFM imaging) as detailed in Fig. 12, among many others (Tapping Mode is a commercialized name for Amplitude Mode also known as Intermittent Contact, Semi-contact, Alternative/alternating current Mode). It provides only topographic images whereas BMR NM provides topographic images as well as nanomechanical maps, simultaneously and in a never achieved before broad range of moduli and bandwidth. There are few publications like the one shared by the reviewer^[5] which attempted to combine the MAC with other modes to shift it to second box, yet they remain qualitative.

We hope that we clearly showed the difference between the MAC and this work. We respect,

Figure 12: Detailed road map to Magnetic Alternative Current (MAC) mode AFM imaging method

acknowledge and appreciate the hard work put into earlier studies. To show this, we added a historical development section in the Supplementary Information. Moreover, we have changed the sentence: **“As shown on the left of Figure 1a, the magnetic-drive peak force modulation AFM is composed of two main components:”** to **“The experimental setup for BMR NM consists of the most advantageous parts of its predecessors (see SI), combined with our novel approaches. Two main components, similar to prior magnetic-actuation setups³⁵⁻³⁷ can be seen in Figure 1a...”**, with suitable references including the MAC work.

2) The second issue is with regards to comparison. While not many people have access to MAC mode instruments and asking for comparison in imaging quality with MAC mode might be unfair, the authors ought at least to show some examples of how the sample they used appear when analysed with the now state-of-the-art Peak Force QNM (Bruker/Veeco). For instance, the granular surface apparent in Fig 3A would that also show up with carefully adjusted settings in Peak Force QNM? Or for that matter (if access allows) in MAC mode AFM?

■ **Reply:** The reviewer brings up an essential point in regards to the comparison of our instrument’s capabilities against all others. Currently there are several nanomechanical mapping instruments available on the market: such as Bruker’s PeakForce QNM, JPK’s QI, Asylum Research’s Bludrive etc. Among all, Bruker is the most transparent company in terms of their instruments’ specifications. They are also the one providing the most resource to public. Which is something they often bring up in meetings as a means to show how confident they are in leading the market. Although other companies are not as transparent, from the publications using their instruments, we can see that they are almost the same caliber. Due to these reasons, we, like the reviewer, consider the Bruker’s PeakForce QNM as the best representative of nanomechanical mapping. It is then no coincidence that we used Bruker’s PeakForce QNM calibration samples to show-case our instrument’s abilities. These samples are well known and studied by the researchers and industry. Examples of which from The Peakforce QNM User Guide can be found on the web:

http://www.torontomicrofluidics.ca/cms/manuals/peak_force.pdf provides images for all the Bruker's PeakForce QNM samples. Below are two of image samples of hard PDMS and HOPG taken in this manual.

Judging by these modulus images taken by the Bruker's own engineers, one can clearly see that the PDMS 2 sample would show similar granular structure as seen in Fig. 13a (ergo, also in the height image) of our manuscript. Likewise, HOPG sample shares a striking similarity to Fig. 13b. We believe ours at least have the equal quality with the image above.

Figure 13: Elastic modulus images of Hard PDMS and HOPG samples from Bruker's calibration kit. Images are taken from http://www.torontomicrofluidics.ca/cms/manuals/peak_force.pdf

In addition, in the section 2.3 (page 5) of the Bruker Peakforce QNM User Guide (http://www.torontomicrofluidics.ca/cms/manuals/peak_force.pdf) gives suggestions for probe selection: **“It is important to choose a probe that can cause enough deformation of the sample and still retain high force sensitivity. Therefore cantilever stiffness should be selected based on the sample stiffness. Brukers recommendations are shown in Table 2.3a.”** From this table, four probes (0.5, 5, 40 and 200 N/m) are recommended to cover the elastic modulus range 1 MPa to 20 GPa. In contrast, our method can cover this range with only one soft probe (2 N/m) because of three characteristics: (i) direct cantilever excitation with magnetic drive at off-resonance frequencies and sufficient indentation forces for broad nanomechanical measurement, (ii) direct force and indentation depth measurement during force-distance curve acquisition, (iii) precise nonlinearity compensation of the optical lever and the fully considered probe damping and inertial effects. With this approach, one can successfully map nanomechanical properties of different samples with elastic moduli ranging over four orders of magnitude using a single probe with high-resolution, both in liquid and air. In particular, to our knowledge, the ability to nanomechanically map heterogeneous surfaces with large elastic modulus variations is reported for the first time using the regulated peak force modulation AFM approach.

At this point, it must be made clear that current work does not attempt to show superiority in terms of imaging resolution but in terms of unifying the modulus spectra for nanomechanical mapping. Almost all currently available AFM systems share similar resolution specifications (Sub-angstrom for vertical, and $\sim 4\text{--}40$ nm for lateral resolution) which are sufficient for the most in demand applications in biology, medicine, material science etc. Our resolution is well within these limits and

maybe not as good but still comparable to the best in the market, as can be seen in the Figures of the manuscript. We would neither attempt to exceed these limits as in a competition with billion dollar companies which employ numerous outstandingly talented engineers and scientists to make their products as compact, stable and noise-free as possible nor it has been our goal. Our home-made product is built modular and not nearly as well vibration-isolated as any high-end products in the market, to demonstrate the capability of our novel approaches to broaden the modulus measurement range of single probes.

References

- (1) Martin, Y. & Wickramasinghe, H. K. Magnetic Imaging by Force Microscopy with 1000-Å Resolution. *Appl Phys Lett*, 1987, 50, 1455-1457, doi:Doi 10.1063/1.97800.
- (2) Marti, O. et al. Mechanical and Thermal Effects of Laser Irradiation on Force Microscope Cantilevers. *Ultramicroscopy*, 1992, 42, 345-350, doi:Doi 10.1016/0304-3991(92)90290-Z.
- (3) Han, W., Lindsay, S. & Jing, T. A magnetically driven oscillating probe microscope for operation in liquids. *Appl Phys Lett*, 1996, 69, 4111–4113.
- (4) Schemmel, A. & Gaub, H. Single molecule force spectrometer with magnetic force control and inductive detection. *Review of scientific instruments*, 1999, 70, 1313–1317.
- (5) Schindler, H. et al. Optimal sensitivity for molecular recognition MAC-mode AFM. *Ultramicroscopy*, 2000, 82, 227–235, doi:Doi 10.1016/S0304–3991(99)00146-1.
- (6) Ge, G; Han, D., Lin, D., Chua, W.; Sun, Y., Jiang, L., Ma, W., & Wang, C. MAC mode atomic force microscopy studies of living samples, ranging from cells to fresh tissue. *Ultramicroscopy*, 2007, 107(4): 299–307.

Reviewers' Comments:

Reviewer #1:

Remarks to the Author:

The revised Ms. has been changed to address some of the reviewers' comments. The authors provide detailed rebuttal letter.

Based on those grounds I recommend the Ms. for publication in NCOMMS after the following comment is addressed:

1 In the revised version of the conclusion is stated ' the ability to nanomechanically map heterogeneous surfaces with large elastic modulus variations is reported for the first time'.

Actually the above statement is not correct. Ref. 34 has the same claim.

Reviewer #2:

Remarks to the Author:

I have read the revised version of this manuscript, and carefully considered the points raised also by the other referees.

Overall, the manuscript has been improved and additional biological samples been measured. Nonetheless, I remain highly skeptical about the novelty of this work: On the one hand, the idea of a magnetic-drive AFM has been around for a long time, as noted by referee 3; on the other hand, the approach brings a moderate advantage in comparison to the well established PF-QNM, since only 2 cantilevers with different spring constants are needed to cover the same range of the Young's moduli. This seems, however, a very limited achievement and a moderate level of novelty to make this work appearing in Nature Communications. To my own understanding, this is more of an incremental work, which I am sure would find its merit in a more focused/technical journal.

Reviewer #3:

Remarks to the Author:

I am very pleased to see the Authors response to my concern and how well they articulated the difference between their method and the old-style magnetic-based methods such as MAC mode. The previous method did not make this clear enough but the new version does, both in the main paper and by the inclusion of the historical perspective in the SI that I found very useful.

To put it more simply, now it is a lot clearer that the method here is the next generation of AFM nanomechanical mapping and in all likelihood the biggest change we seen in the AFM field since the Peak Force QNM hit the market (followed by various similar systems from JPK, Asylum etc.. – I also agree with the authors that Bruker thankfully have been quite transparent on how their instruments work).

The additional explanations and biological samples provided in response to the other reviewers also add value to the paper.

Hence, while I was a bit sceptical before, I now totally agree with the authors that this manuscript should be published and I think it could have a big impact. I would be rather surprised if the commercial AFM vendors don't take this one.

Just one very minor comment (and I don't need to review the revised manuscript to see the answer): Currently the authors put their lower limit in terms of mechanical resolution around 1 kPa in liquids. That's impressive however, some neurons at least as softer yet. Could the authors

comment on in the conclusions how (if theoretically possible), they could see future developments pushing their methods towards 0.1 kPa or even softer materials in liquid?

1. Response to comments from Reviewer #1

Dear Reviewer,

Please see our statements to your valuable comments on our manuscript:

I. Comments:

The revised Ms. has been changed to address some of the reviewers' comments. The authors provide detailed rebuttal letter. Based on those grounds I recommend the Ms. for publication in NCOMMS after the following comment is addressed:

1 In the revised version of the conclusion is stated ' the ability to nanomechanically map heterogeneous surfaces with large elastic modulus variations is reported for the first time'.

Actually the above statement is not correct. Ref. 34 has the same claim.

- **Reply:** We thank the reviewer for the recommendation and understand the concern. In the previous revised version, we stated " the ability to nanomechanically map heterogeneous surfaces with large elastic modulus variations is reported for the first time using the regulated peak force modulation AFM approach." This sentence is intended to express that it is the first time to applied **regulated** peak force on deform portion of the sample according to its elastic modulus with the peak force modulation AFM approach. As the reviewer mentions, Ref. 34 is the first time to show the ability to nanomechanically map heterogeneous surfaces with large elastic modulus. However, they use torsional harmonic tapping AFM not peak force modulation AFM which is based on force-distance curves to measure material nanomechanics. In addition, torsional harmonic tapping AFM didn't regulate the peak force during the process of nanomechanical mapping.

Based on reviewer's comment, we consider that this expression is easy to be misunderstood. In addition, NCOMMS is a highly selective journal so the novelty of the work will be clear to the reader. Therefore, We have **deleted** this sentence in the revised version.

2. Response to comments from Reviewer #2

Dear Reviewer,

Please see our statements to your valuable comments on our manuscript:

I. Comments:

I have read the revised version of this manuscript, and carefully considered the points raised also by the other referees.

Overall, the manuscript has been improved and additional biological samples been measured. Nonetheless, I remain highly skeptical about the novelty of this work: On the one hand, the idea of a magnetic-drive AFM has been around for a long time, as noted by referee 3; on the other hand, the approach brings a moderate advantage in comparison to the well established PF-QNM, since only 2 cantilevers with different spring constants are needed to cover the same range of the Young's moduli. This seems, however, a very limited achievement and a moderate level of novelty to make this work appearing in Nature Communications. To my own understanding, this is more of an incremental work, which I am sure would find its merit in a more focused/technical journal.

■ **Reply:** We thank the referee for the comment and wish to express the disappointment we feel for not being able to satisfy his/her intellectual needs. In our humble opinion, invention and innovation require creative, original ways to use the accumulated information to generate more of it and/or ways to use it. Just because the idea of a concept has been out there for a long time, its innovations or their impact should not be underestimated. The wheel that was invented in Neolithic times is undoubtedly not the same (or insignificantly different) as its modern forms. With the additional information provided in the previous revision, we succeeded in showing the novelty and difference of our system compared to all other methods to the satisfaction of referee 3, who states: "I am very pleased to see the Authors response to my concern and how well they articulated the difference between their method and the old-style magnetic-based methods such as MAC mode. ... To put it more simply, now it is a lot clearer that the method here is the next generation of AFM nanomechanical mapping and in all likelihood the biggest change we seen in the AFM field since the Peak Force QNM hit the market. ... Hence, while I was a bit sceptical before, I now totally agree with the authors that this manuscript should be published and I think it could have a big impact. I would be rather surprised if the commercial AFM vendors don't take this one." In case, our reply to referee 3 has not been sent to you, we would like to share it below.

As for the advantage over PF-QNM that is deemed to be moderate, we would like to remind the referee that this work not only provides scientific innovation and the biggest step towards completely unified elasticity spectrum so far, but also cost efficiency. The efficiency appears in the time consumed for the change and calibration of the cantilever as well as the cost of buying various cantilevers. Nevertheless, we respect the referees opinion and hope that it will change with the impactful future works citing this one.

Reply to referee 3 in the previous rebuttal letter:

Like the reviewer references, magnetically driven AFM probes and their applications have been reported in the literature for a longtime. After the introduction of Magnetic Force Microscopy (MFM) in 1987 by Martin and Wickramasinghe^[1] it was realized that the magnetic forces can be used directly to bend the cantilever. Especially after the success of other direct probe actuation methods such as photothermal heating in 1992 by Marti et al.^[2] many researchers have been using magnetic cantilevers either by coating them with magnetic materials such as cobalt via thin film

deposition methods or gluing small magnets to the backside of the cantilever. Thus, their use was extended into AC imaging such as in the article^[3], the reviewer was kind to share with us. We would like to remind that MAC is an imaging-only method. From this point onward MAC method and the likes have been employed for direct probe control for imaging, until in 1998 Schemmel and Gaub introduced single molecule force spectrometer with magnetic force control and inductive detection^[4]. With this development magnetically coated or magnetic particle decorated cantilevers were driven for force spectroscopy either with the force of a magnetic field gradient or the torque of any perpendicular field. We would like to point out that all these methods were employed for conventional force spectroscopy in contrast to quantitative force mapping which is the most advanced version of force spectroscopy, then nanomechanical mapping which is the most advanced version of quantitative force mapping. In our study, we used magnetic torque driven cantilevers for the most advanced version of nanomechanical mapping. For this reason, as well as the length constraints of a communication article, we limited our discussion to nanomechanical mapping modes hence cited related articles.

Moreover, as shown in Figure 1, the magnetic bead in our method is placed directly on top the tip by precision micromanipulation technique, unlike magnetic coatings or glued magnetic particle modifications. This results in the local magnetic excitation of the cantilever, directly aligned along the tip itself as can be seen in the below figure. This way the tip position can be directly controlled. In contrast to magnetic thin film coatings or random shape and position magnetic particles, the dynamics of the exciting force can be accurately followed and remain stable, yielding in constant k value ergo applied force. This becomes, especially important when it comes to the operation in liquid environment.

Figure 1: Configurations of different magnetic drive probes.

We would like to underline that in this paper, we are not focused on the instrumentation but rather our instrument's ability to provide a long sought after answer to the infamous discrete moduli range problem. That is owing to our novel approaches for cantilever calibration, parametric actuation, acquisition, modeling, calculation & visualization of the force-distance data, and mapping

algorithms. In the Fig. 2, we humbly try to show each breakthrough in AFM-based force-distance data handling with oversimplification:

Figure 2: Breakthroughs in AFM-based force-distance data handling.

To summarize, the MAC mode rests within the first box (AFM imaging) as detailed in Fig. 3, among many others (Tapping Mode is a commercialized name for Amplitude Mode also known as Intermittent Contact, Semi-contact, Alternative/alternating current Mode). It provides only topographic images whereas BMR NM provides topographic images as well as nanomechanical maps, simultaneously and in a never achieved before broad range of moduli and bandwidth. There are few publications like the one shared by the reviewer^[5] which attempted to combine the MAC with other modes to shift it to second box, yet they remain qualitative.

Figure 3: Detailed road map to Magnetic Alternative Current (MAC) mode AFM imaging method

Reference

- (1) Martin, Y. & Wickramasinghe, H. K. Magnetic Imaging by Force Microscopy with 1000-Å Resolution. *Appl Phys Lett*, 1987, 50, 1455-1457, doi:Doi 10.1063/1.97800.
- (2) Marti, O. et al. Mechanical and Thermal Effects of Laser Irradiation on Force Microscope Cantilevers. *Ultramicroscopy*, 1992, 42, 345-350, doi:Doi 10.1016/0304-3991(92)90290-Z.
- (3) Han, W., Lindsay, S. & Jing, T. A magnetically driven oscillating probe microscope for operation in liquids. *Appl Phys Lett*, 1996, 69, 4111-4113.
- (4) Schemmel, A. & Gaub, H. Single molecule force spectrometer with magnetic force control and inductive detection. *Review of scientific instruments*, 1999, 70, 1313-1317.
- (5) Schindler, H. et al. Optimal sensitivity for molecular recognition MAC-mode AFM. *Ultramicroscopy*, 2000, 82, 227-235, doi:Doi 10.1016/S0304-3991(99)00146-1.

3. Response to comments from Reviewer #3

Dear Reviewer,

Please see our statements to your valuable comments on our manuscript:

I. Comments:

I am very pleased to see the Authors response to my concern and how well they articulated the difference between their method and the old-style magnetic-based methods such as MAC mode. The previous method did not make this clear enough but the new version does, both in the main paper and by the inclusion of the historical perspective in the SI that I found very useful.

To put it more simply, now it is a lot clearer that the method here is the next generation of AFM nanomechanical mapping and in all likelihood the biggest change we seen in the AFM field since the Peak Force QNM hit the market (followed by various similar systems from JPK, Asylum etc.. I also agree with the authors that Bruker thankfully have been quite transparent on how their instruments work).

The additional explanations and biological samples provided in response to the other reviewers also add value to the paper. Hence, while I was a bit sceptical before, I now totally agree with the authors that this manuscript should be published and I think it could have a big impact. I would be rather surprised if the commercial AFM vendors don't take this one.

Just one very minor comment (and I don't need to review the revised manuscript to see the answer): Currently the authors put their lower limit in terms of mechanical resolution around 1 kPa in liquids. That's impressive however, some neurons at least as soft as yet. Could the authors comment on in the conclusions how (if theoretically possible), they could see future developments pushing their methods towards 0.1 kPa or even softer materials in liquid?

■ **Reply:** We thank the reviewer for positive comments. As the reviewer mentions, the elastic moduli of some neurons and cancer cells are less than 1kPa. It is very meaningful to push our methods towards ultra-soft materials in liquid. In our method, the only factor restrains this application is the probe. Due to the low tip-sample interaction force (less than 1 pN), even the softest commercial AFM probe is not able to provide sufficient force sensitivity. If ultra-soft probes are available in the future, it can be possible to access mechanical resolution below 1 kPa for NM of ultra-soft samples such as neurons. Fortunately, AFM probes made by polymers^[1] are promising to meet such requirements. The relevant content has been added to the revised version.

Reference

- (1) Lee, J. S. et al. Multifunctional hydrogel nano-probes for atomic force microscopy. *Nat. Commun.* **7**, 11566 (2016).